# Scaling properties reveal regulation of river flows in the Amazon through a "forest reservoir"

Juan F. Salazar [1], Juan Camilo Villegas [1,2], Angela M. Rendón [1], Estiven Rodríguez [1], Isabel Hoyos [3,4], Daniel Mercado-Bettín [1], and Germán Poveda [5]

[1]GIGA, Escuela Ambiental, Facultad de Ingeniería, Universidad de Antioquia, Medellín, Colombia
[2]School of Natural Resources and the Environment, University of Arizona, Tucson, USA
[3]GAIA, Escuela Ambiental, Facultad de Ingeniería, Universidad de Antioquia, Medellín, Colombia
[4]Instituto de Física, Universidad de Antioquia, Medellín, Colombia
[5]Universidad Nacional de Colombia, Sede Medellín, Departamento de Geociencias y Medio Ambiente, Facultad de Minas, Medellín, Colombia

*Correspondence to:* Juan F. Salazar (juan.salazar@udea.edu.co)

**Abstract.** Many natural and social phenomena depend on river flow regimes that are being altered by global change. Understanding the mechanisms behind such alterations is crucial for predicting river flow regimes in a changing environment. Here we introduce a novel physical interpretation of the scaling properties of river flows, and show that it leads to a parsimonious characterization of the flow regime of any river basin. This allows to classify river basins as regulated or unregulated, and to identify a critical threshold between these states. We applied this framework to the Amazon river basin and found both states among its main tributaries. Then we introduce the "forest reservoir" hypothesis to describe the natural capacity of river basins to regulate river flows through land-atmosphere interactions (mainly precipitation recycling) that depend strongly on the presence of forests. A critical implication is that forest loss can force the Amazonian river basins from regulated to unregulated states. Our results provide theoretical and applied foundations for predicting hydrological impacts of global change, including the detection of early-warning signals for critical transitions in river basins.

## 1 Introduction

Mean and extreme river flows are global-change-sensitive components of river flow regimes that are determinant for many ecological and societal processes (Zhang et al., 2016; Lima et al., 2014; Sterling et al., 2013; Coe et al., 2009; Piao et al., 2007; Mahe et al., 2005). Landscape and climate alterations foreshadow shifts of precipitation and river flow regimes (Boers et al., 2017; Khanna et al., 2017; Zemp et al., 2017; Lawrence and Vandecar, 2015; Botter et al., 2013; Davidson et al., 2012; Hirota et al., 2011; Sampaio et al., 2007). The conversion of precipitation into river flow through the accumulation of runoff depends on a suite of complex and heterogeneous biophysical processes and attributes of river basins, at different scales (Blöschl et al., 2007; McDonnell et al., 2007). This conversion results in spatial scaling properties —properties that do not vary within a wide range of scales— observable through river flow records (Gupta et al., 2007; Gupta and Waymire, 1990). The existence of scaling properties in river basins implies power law correlation between the system response —river flows— and a scale parameter —typically the drainage area—(Gupta et al., 2007). Power laws go beyond statistical fitting,

they indicate scale-invariance as a fundamental emergent property arising from the self-organization of many complex systems in nature (Kéfi et al., 2007; Sivapalan, 2005; Brown et al., 2002). Scaling properties are common to river basins with very different environmental conditions (Gupta et al., 2010; Poveda et al., 2007). This suggests that the spatial scaling properties of river flows have a common, mechanistic origin, that has been related to conservation principles and the fractal nature of river networks (Gupta et al., 2007; Sivapalan, 2005).

The values of the scaling parameters —the scaling exponent and coefficient of a given power law— are neither universal nor static features of river basins, because they depend on runoff production processes that are spatially heterogeneous (Blöschl et al., 2007; McDonnell et al., 2007) and sensitive to both climate and land cover change (Sterling et al., 2013; Coe et al., 2009; Piao et al., 2007; Mahe et al., 2005). Understanding the mechanisms behind the scaling parameters in river basins, as well as their sensitivity to global change, is a crucial step for enabling the use of the scaling theory in hydrological *prediction in ungaged basins* (the "PUB problem"; Hrachowitz et al., 2013) and, more generally, in a changing environment where the processes governing the hydrological cycle are not static (the "Panta Rhei—Everything Flows" debate; Montanari et al., 2013). We address this problem by linking the scaling properties of river flows to the capacity of river basins for regulating their hydrological response.

## 2   Scaling Properties Reveal River Flow Regulation

The scaling properties of river flows are evidenced through power laws of the form (Gupta and Waymire, 1990)

$$E[Q_i^k] = \alpha_i S^{\beta_i},$$
(1)

where $E[Q_i^k]$ is the $k$th order statistical moment of the probability distribution function of river flows, $S$ is a scale parameter, and $\alpha_i$ and $\beta_i$ are the scaling coefficient and exponent, respectively. $Q_i$ can be floods ($i = F$), mean flows ($i = M$) or low flows ($i = L$). The scaling parameters ($\alpha_i$ and $\beta_i$) vary among river basins and flow types, and are always positive because river flows cannot be negative and increase downstream as a consequence of mass continuity.

The state of a river basin can be classified as *regulated* or *unregulated* depending on its river flow regime, which determines how the scaling exponents for floods ($\beta_F$), mean flows ($\beta_M$) and low flows ($\beta_L$) are organized. Regulation is defined here as the capacity of river basins to attenuate the amplitude of the river flow regime, that is to reduce the difference between floods and low flows. A river basin is regulated if $\beta_L > \beta_M > \beta_F$ or unregulated if $\beta_L < \beta_M < \beta_F$. A metric of the extremes amplitude is the difference ($\Delta_Q$) between long-term average floods ($E[Q_F]$) and low flows ($E[Q_L]$), relative to mean flows ($E[Q_M]$),

$$\Delta_Q = \frac{E[Q_F] - E[Q_L]}{E[Q_M]} = \frac{\alpha_F S^{\beta_F} - \alpha_L S^{\beta_L}}{\alpha_M S^{\beta_M}}.$$
(2)

Our distinction between regulated and unregulated states is consistent with the definition of regulation in artificial reservoirs, whereby a reservoir regulates river flows by either mitigating floods through water retention or enhancing low flows through

water release (Magilligan and Nislow, 2005). The extremes amplitude is dampened in the regulated state ($\Delta_Q$ is reduced as $S$ increases), or amplified in the unregulated state ($\Delta_Q$ is increased as $S$ increases), as a consequence of how river flows grow downstream in a river basin. These contrasting behaviours are reflected by the scaling exponents through the spatial rate of change

$$\frac{\partial \Delta_Q}{\partial S} = \frac{\alpha_F S^{\beta_F}(\beta_F - \beta_M) + \alpha_L S^{\beta_L}(\beta_M - \beta_L)}{\alpha_M S^{\beta_M + 1}} \begin{cases} < 0, & \text{if } \beta_L > \beta_M > \beta_F \text{ (regulated state)} \\ = 0, & \text{if } \beta_L = \beta_M = \beta_F \text{ (critical threshold)} \\ > 0, & \text{if } \beta_L < \beta_M < \beta_F \text{ (unregulated state)}. \end{cases}$$
(3)

The difference between the regulated and unregulated states is evidenced by the theoretical limit

$$\lim_{S \to \infty} \Delta_Q = \begin{cases} 0, & \text{if } \beta_L > \beta_M > \beta_F \text{ (regulated state)} \\ (\alpha_F - \alpha_L)/\alpha_M \text{ (a positive constant)}, & \text{if } \beta_L = \beta_M = \beta_F \text{ (critical threshold)} \\ \infty, & \text{if } \beta_L < \beta_M < \beta_F \text{ (unregulated state)}. \end{cases}$$
(4)

In the regulated state, the flow regime tends to the limit of complete regulation (constant flow: $E[Q_F] = E[Q_M] = E[Q_L]$), owing to the capacity of the river basin to dampen extremes ($\Delta_Q \to 0$). The opposite occurs in the unregulated state: the
extremes are amplified ($\Delta_Q \to \infty$) and, hence, $E[Q_F] >> E[Q_M] >> E[Q_L]$. Therefore, in a given river basin, reversing the direction of the inequality from $\beta_L > \beta_M > \beta_F$ to $\beta_L < \beta_M < \beta_F$ indicates a shift between the regulated and unregulated states, with $\beta_L = \beta_M = \beta_F$ being a critical threshold. This agrees with the definition of a tipping point as "*the corresponding critical point —in forcing and a feature of the system— at which the future state of the system is qualitatively altered*" (Lenton, 2011). The difference ($\beta_L - \beta_F$) denotes a metric of the regulation level that indicates the proximity to the critical threshold in
a river basin. Everything else being equal, a reduction of $\beta_L$ indicates an increased severity of low flows, whereas an increase of $\beta_F$ indicates an increase of flood severity.

   The occurrence of regulated or unregulated states depends on the combined effect of *dampening* and *amplification* processes operating within a river basin. Both processes can coexist in a regulated river basin because higher regulation implies both reducing floods through a dampening effect produced by water retention within the basin, and increasing low flows through
an amplification effect resulting from the release of water stored within the basin. The occurrence of either of these effects is described by how the rate of change

$$\frac{\partial E[Q_i^k]}{\partial S} = \alpha_i \beta_i S^{\beta_i - 1}$$
(5)

grows with increasing scale. If $\partial E[Q_i^k]/dS$ decreases with $S$ —power law (1) is convex in $S$—, then the flows are dampened within the river basin, meaning that the production of runoff per unit area decreases downstream along the river network.

The opposite occurs if $\partial E[Q_i^k]/dS$ increases with $S$ —power law (1) is concave in $S$—. Whether $\partial E[Q_i^k]/dS$ increases or decreases with increasing $S$ is determined by the value of the scaling exponent $\beta_i$ relative to 1, as given by

$$\frac{\partial^2 E[Q_i^k]}{\partial S^2} = \alpha_i \beta_i (\beta_i - 1) S^{\beta_i - 2} \begin{cases} < 0, & \text{if } 0 < \beta_i < 1 \text{ (dampening process)} \\ = 0, & \text{if } \beta_i = 1 \text{ (critical point)} \\ > 0, & \text{if } \beta_i > 1 \text{ (amplification process)}, \end{cases} \tag{6}$$

whereby $0 < \beta_i < 1$ and $\beta_i > 1$ represent, respectively, the dampening and amplification processes, and $\beta_i = 1$ is a critical value around which the curvature of power law (1) —and therefore the sign of its second derivative— changes. Higher regulation leads to dampened floods ($0 < \beta_F < 1$) and enhanced low flows ($\beta_L > 1$).

## 3  Regulated and Unregulated Basins in the Amazon

We tested our physical interpretation of the scaling properties in the Amazon river basin as a whole, and in its major sub-basins treated as independent systems (Fig. 1). Large-scale forest degradation or loss is a major driver of environmental change in these river basins (Boers et al., 2017; Khanna et al., 2017; Zemp et al., 2017; Lawrence and Vandecar, 2015; Lima et al., 2014; Davidson et al., 2012; Hirota et al., 2011; Coe et al., 2009). The capacity to maintain high evapotranspiration rates is a key attribute of Amazonian forests associated with their large cumulative area of leaves (Caldararu et al., 2012; von Randow et al., 2012; Da Rocha et al., 2009). We take this into account by setting the scaling parameter as $S = LA = A \times \overline{LAI}$, where $\overline{LAI}$ is the leaf area index averaged over the drainage area $A$ of each basin, so power law (1) becomes

$$E[Q_i^k] = \alpha_i LA^{\beta_i}. \tag{7}$$

We tested the consistency of our results when using $E[Q_i^k] = \gamma_i A^{\delta_i}$ instead of Eq. (7), i.e. by setting $A$ as the scale parameter (results are included in the Supplementary Information). Using basin topographic data and daily river flow records from 85 gauges from the SO-HYBAM project (Cochonneau et al., 2006, Fig. 1 and Supplementary Table S1), and $LAI$ data (Liu et al., 2012) averaged for 1981–2012 (Fig. 1), we found that annual mean and extreme river flows ($E[Q_i^k]$ with $k = 1$) in the Amazonian basins exhibit significant ($p < 0.05$, $t$-test results are in Supplementary Table S2) scaling properties through power laws of the form (7) (Fig. 2). Likewise, the scaling properties are evident when using $A$ as the scale parameter (Supplementary Figures S1–S7 and Table S9).

Estimated values of the scaling exponents reveal the existence of both regulated and unregulated basins within the Amazon (Fig. 3 and Supplementary Fig. S8). The Amazon, Negro, Solimoes and Madeira river basins are regulated as indicated by their scaling exponents: $\beta_L > \beta_M > \beta_F$ —statistical significance of the comparisons between the scaling exponents in the Xingu river is limited because of the few degrees of freedom determined by the number of gauges, so we excluded this basin from this analysis—. In these regulated basins, $\Delta_Q$ decreases with the spatial scale, as given by (3) and (4) with $\beta_L > \beta_M > \beta_F$

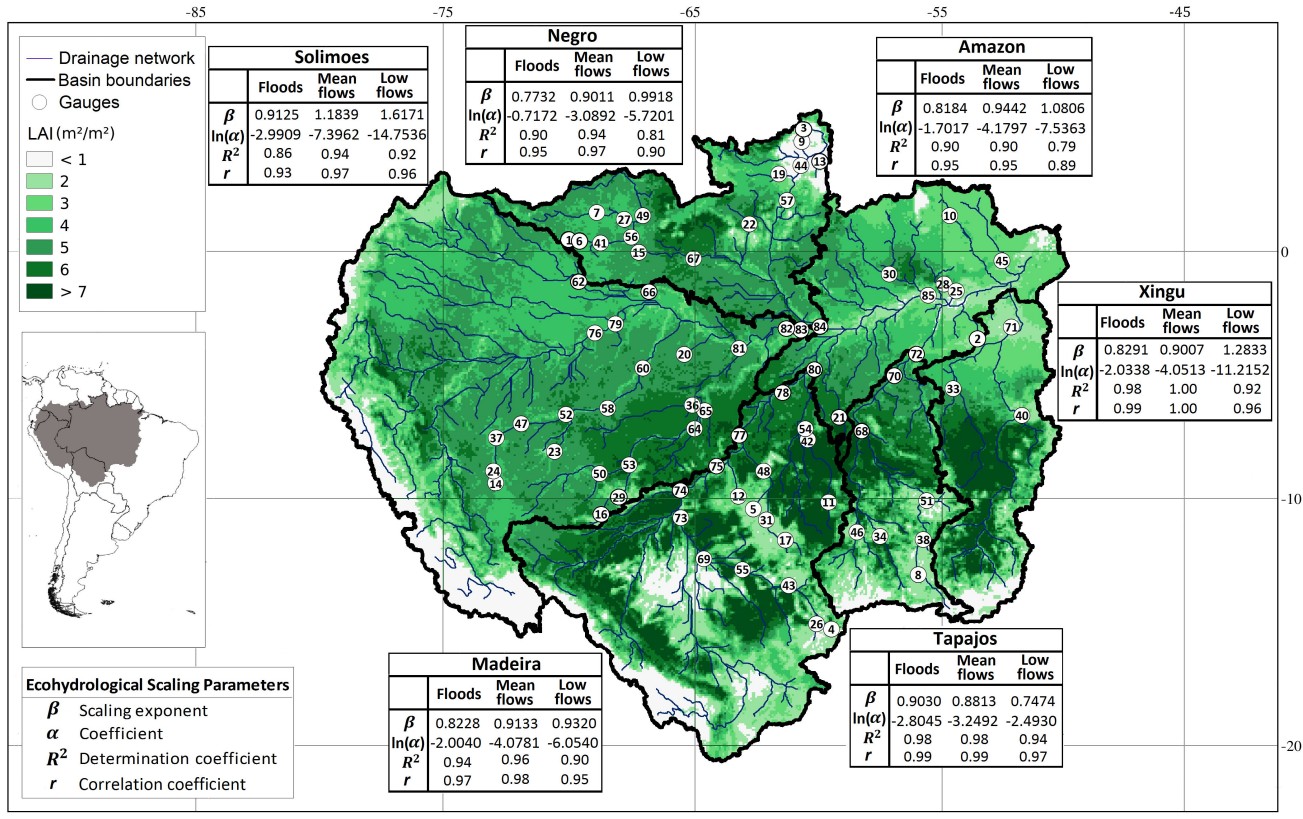

**Figure 1.** The Amazon basin and its major sub-basins. The map shows the long-term leaf area index averaged over the period 1981–2012, boundaries and drainage network of the sub-basins, and river flow gauges provided by the SO-HYBAM project (www.ore-hybam.org). Detailed information about the gauges is in Supplementary Table S1. Tables show the parameters of power laws for mean and extreme river flows in each basin.

(Fig. 4 and Supplementary Fig. S9). In contrast, the scaling exponents ($\beta_L < \beta_M < \beta_F$) indicate that the Tapajos river basin has already transitioned into the unregulated state, whereby $\Delta_Q$ is not reduced with the spatial scale (Fig. 4f).

River basins can be classified by their regulation level: $\beta_L - \beta_F$ (Table 1). The Solimoes is the more regulated basin ($\beta_L - \beta_F = 0.70 > 0$), while the Madeira is still regulated but close to the critical threshold ($\beta_L - \beta_F = 0.11 > 0$) and the Tapajos basin has already transitioned into the unregulated state ($\beta_L - \beta_F = -0.16 < 0$). The Amazon as a whole is in the regulated state, but it is less regulated than the Solimoes ($\beta_L - \beta_F = 0.26$), consistent with the presence of the less regulated basins within the whole Amazon. In the following section, we explore the physical mechanisms behind the occurrence of different regulation states.

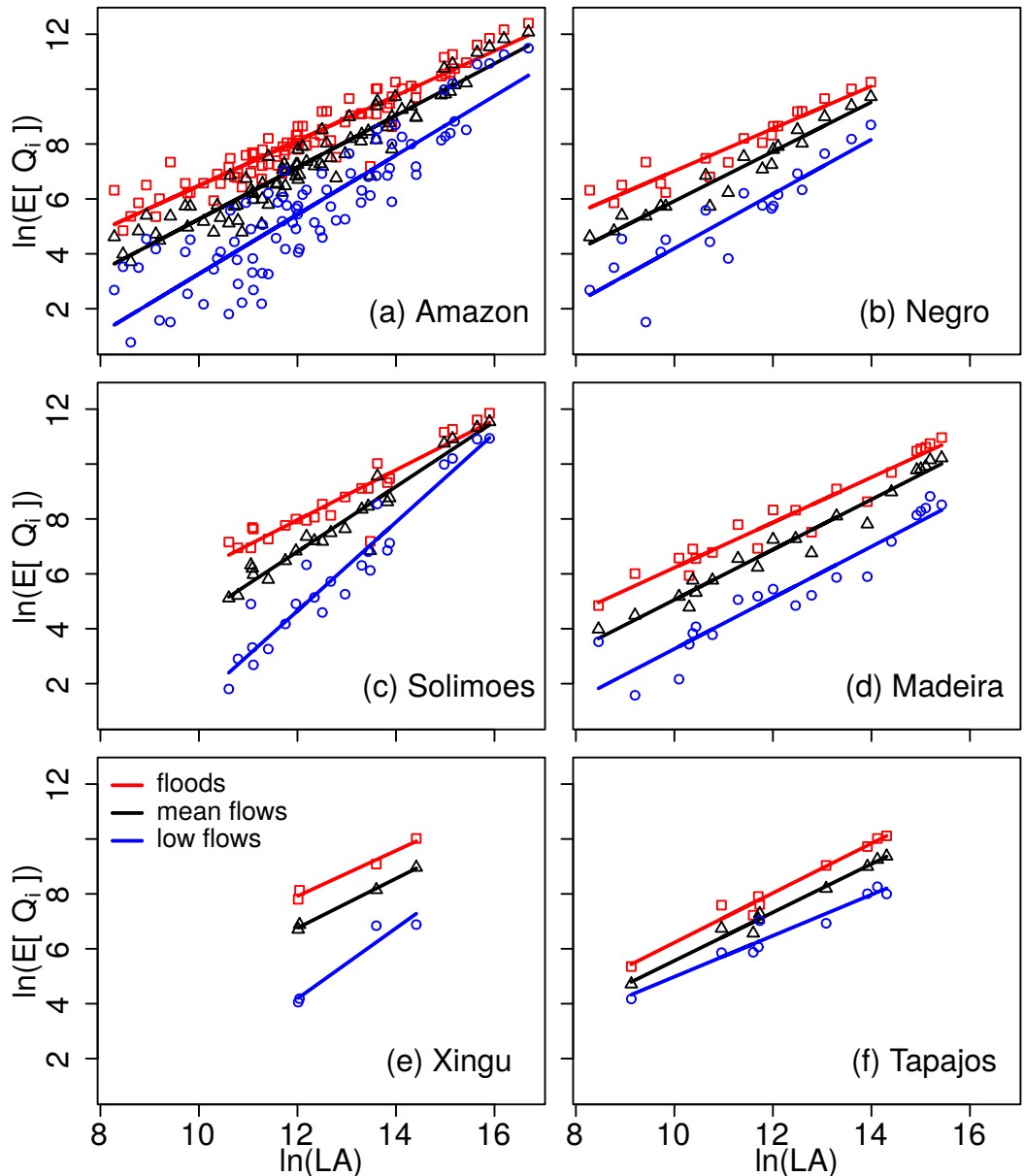

**Figure 2.** Power laws of the form $E[Q_i] = \alpha_i LA^{\beta_i}$ (Eq. 7 with $k = 1$) for low flows ($i = L$), mean flows ($i = M$), and floods ($i = F$). Points are observed river flows and lines are the scaling relations (in all cases $r > 0.88$ and $p < 0.05$). **(a) Amazon:** $E[Q_L] = \exp(-7.53)LA^{1.08}$; $E[Q_M] = \exp(-4.18)LA^{0.94}$; $E[Q_F] = \exp(-1.70)LA^{0.82}$. **(b) Negro:** $E[Q_L] = \exp(-5.72)LA^{0.99}$; $E[Q_M] = \exp(-3.09)LA^{0.90}$; $E[Q_F] = \exp(-0.71)LA^{0.77}$. **(c) Solimoes:** $E[Q_L] = \exp(-14.75)LA^{1.62}$; $E[Q_M] = \exp(-7.40)LA^{1.18}$; $E[Q_F] = \exp(-2.99)LA^{0.91}$. **(d) Madeira:** $E[Q_L] = \exp(-6.05)LA^{0.93}$; $E[Q_M] = \exp(-4.08)LA^{0.91}$; $E[Q_F] = \exp(-2.00)LA^{0.82}$. **(e) Xingu:** $E[Q_L] = \exp(-11.22)LA^{1.28}$; $E[Q_M] = \exp(-4.05)LA^{0.90}$; $E[Q_F] = \exp(-2.03)LA^{0.83}$. **(f) Tapajos:** $E[Q_L] = \exp(-2.49)LA^{0.75}$; $E[Q_M] = \exp(-3.25)LA^{0.88}$; $E[Q_F] = \exp(-2.80)LA^{0.90}$. For convenience, $\alpha_i$ is expressed as $\exp(\ln(\alpha_i))$.

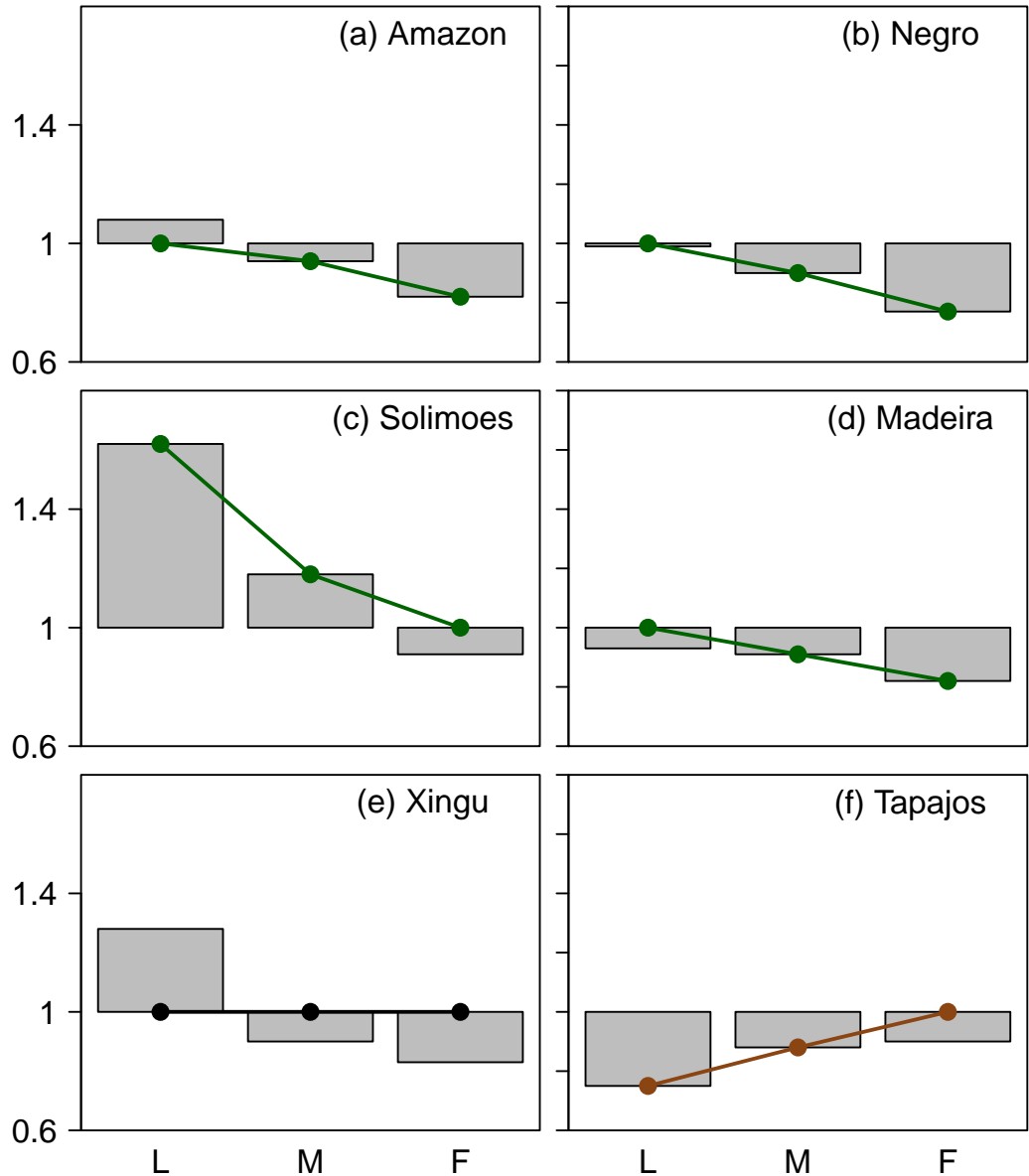

**Figure 3.** Observed patterns of the values of the scaling exponents ($\beta_i$) for low flows (L), mean flows (M), and floods (F), in the Amazon basin and its six major sub-basins. Dots over the bars indicate whether the scaling exponent is significantly different to 1 ($p < 0.05$, the dot is not over 1) or not (the dot is over 1). Details about the $t$-tests are in Supplementary Tables S3 to S8. In regulated states (green, a–d), the exponents decrease from low flows to floods; whereas in unregulated states (brown, f), the exponents increase from low flows to floods. In the Xingu river basin (e), the hypothesis that all exponents are equal to 1 cannot be rejected ($p > 0.05$) because of the small number of degrees of freedom (gauges).

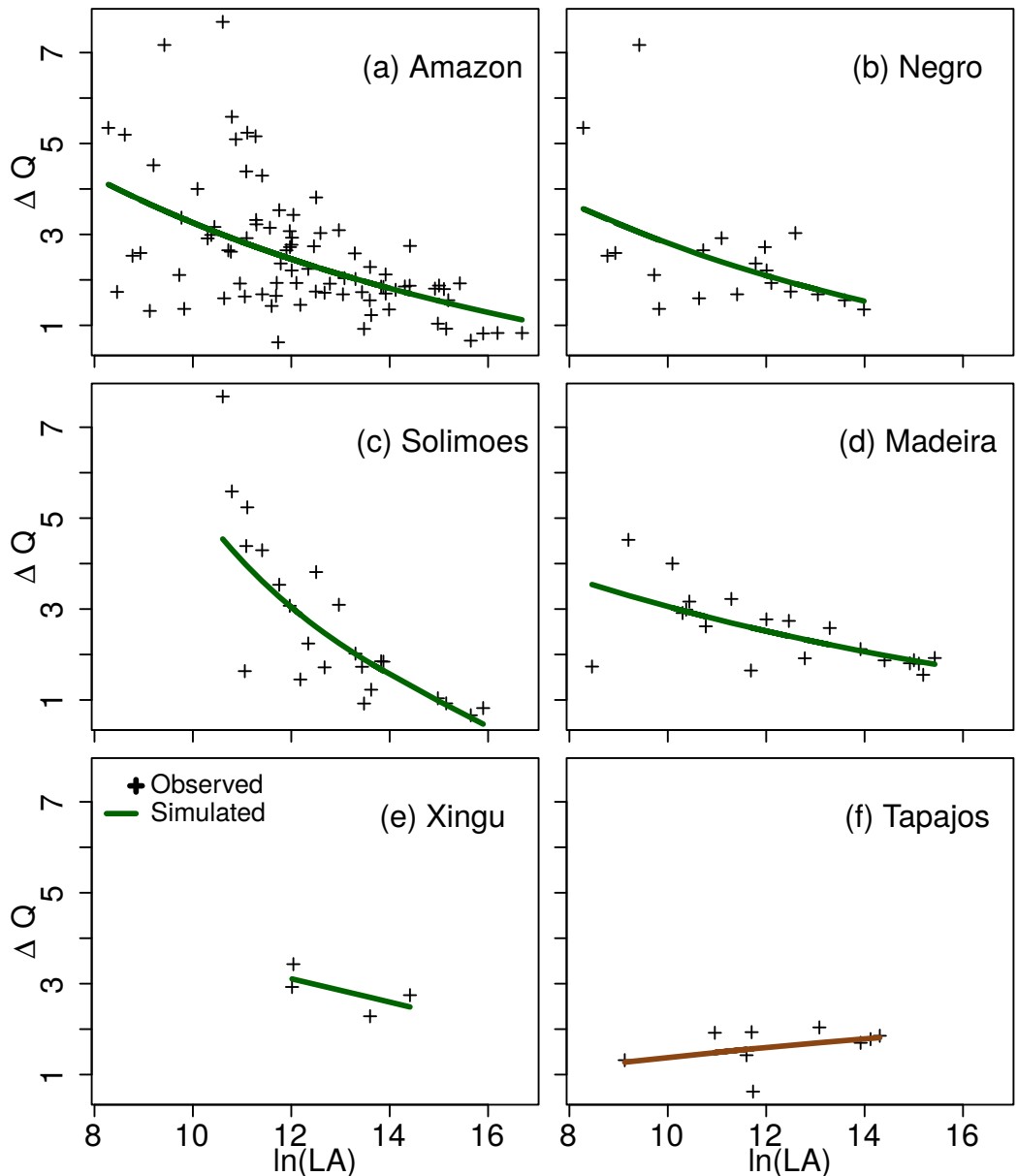

**Figure 4.** Extremes amplitude, $\Delta_Q = (E[Q_F] - E[Q_L])/E[Q_M]$, as observed (crosses) and simulated (lines) by $(\alpha_F LA^{\beta_F} - \alpha_L LA^{\beta_L})/\alpha_M LA^{\beta_M}$ (from Eq. 2 with $S = LA$), using the scaling parameters of each basin. $\Delta_Q$ either decreases or increases with spatial scale ($LA$) depending on whether the river basin is regulated ($\beta_L > \beta_M > \beta_F$, e.g. Solimoes) or unregulated ($\beta_L < \beta_M < \beta_F$, e.g. Tapajos).

**Table 1.** River flow regulation state and level in each basin as revealed by the scaling exponents of power laws $E[Q_i] = \alpha_i LA^{\beta_i}$ (or $E[Q_i] = \gamma_i A^{\delta_i}$). Difference $\beta_L - \beta_F$ (or $\delta_L - \delta_F$) indicates both the regulation state (regulated if positive, unregulated if negative) and the proximity to the critical threshold or regulation level (magnitude of the difference). Basins are ordered from top to bottom by their regulation level.

| River basin | $\beta_L - \beta_F$ ($\delta_L - \delta_F$) | State | Behaviour of the extremes with increasing spatial scale ($LA$ or $A$) |
|---|---|---|---|
| Solimoes | 0.70 (0.67) | Regulated | The extremes amplitude ($\Delta_Q$) is greatly reduced (Fig. 4c and Supplementary Fig. S9c) because of a strong capacity of the basin for amplifying low flows ($\beta_L = 1.62 >> 1.00$ and $\delta_L = 1.55 >> 1.00$) while not amplifying floods ($\beta_F = 0.91 \leq 1.00$ and $\delta_F = 0.88 \leq 1.00$). |
| Amazon | 0.26 (0.31) | Regulated | $\Delta_Q$ is reduced (Fig. 4a and Supplementary Fig. S9a) due to the combined effect of low flows amplification ($\beta_L = 1.08 \geq 1.00$ and $\delta_L = 1.17 > 1.00$) and floods dampening ($\beta_F = 0.82 < 1.00$ and $\delta_F = 0.86 < 1.00$). |
| Negro | 0.22 (0.17) | Regulated | $\Delta_Q$ is reduced (Fig. 4b and Supplementary Fig. S9b) because of the basin's capacity for dampening floods ($\beta_F = 0.77 < 1.00$ and $\delta_F = 0.90 < 1.00$) while not dampening low flows. Low flows grow approximately linearly with scale ($\beta_L = 0.99 \approx 1.00$ and $\delta_L = 1.07 \geq 1.00$). |
| Madeira | 0.11 (0.14) | Regulated | $\Delta_Q$ is reduced (Fig. 4d and Supplementary Fig. S9d) mainly because of the basin's capacity for dampening floods ($\beta_F = 0.82 < 1.00$ and $\delta_F = 0.86 < 1.00$). Low flows are not amplified ($\beta_L = 0.93 \leq 1.00$ and $\delta_L \approx 1.00$). |
| Tapajos | -0.16 (-0.20) | Unregulated | $\Delta_Q$ is increased (Fig. 4f and Supplementary Fig. S9f) because low flows are not amplified ($\beta_L = 0.75 < 1.00$, $\delta_L = 0.89 \leq 1.00$) and floods are less dampened than low flows ($1.00 \geq \beta_F = 0.90 > \beta_L = 0.75$) or even amplified ($\delta_F = 1.09 > 1.00 \geq \delta_L = 0.89$). |

## 4 Discussion

### 4.1 The use of $LA$ as scale parameter

Our general idea about the classification of river basins is independent of using $LA$ as the scale parameter. The interpretation of the scaling properties presented in Section 2 is based only on the assumption that river flows in a given river basin exhibit scaling properties through power laws of the form of Eq. (1). This does not require the use of $LA$ as the scale parameter. Instead, it allows to investigate the use of different scale parameters (e.g. Poveda et al., 2007): all of the equations in Section 2 use $S$ as a general scale parameter that could be replaced by different factors depending on the case study. $LA$ was introduced as the scale parameter for the application of our general framework (Section 2) to the particular case of the Amazon (Section 3). The idea is not that $LA$ must be used as the scale parameter in any river basin, but to show that it can be successfully used as a scale parameter in the Amazon.

Although using $LA$ as the scale parameter does not always improve $R^2$ in the scaling power laws (Supplementary Figs. S1–S6), the main results of our study are statistically significant and consistent among the two scaling models: $E[Q_i^k] = \alpha_i LA^{\beta_i}$ (using $LA$) and $E[Q_i] = \gamma_i A^{\delta_i}$ (using $A$). Both models agree in the ordering of basins by their regulation level, and that the Tapajos basin is unregulated (Table 1). The most conspicuous difference between the models is that they do not fully agree in the description of amplifying and dampening processes in the Tapajos basin (Table 1). However, both models agree that, in this basin: (i) low flows are not amplified and can even be dampened ($\beta_L = 0.75 < 1.00$; $\delta_L = 0.89 \leq 1.00$); and (ii) floods are less dampened than low flows ($1.00 \geq \beta_F = 0.90 > \beta_L = 0.75$) or even amplified ($\delta_F = 1.09 > 1.00 \geq \delta_L = 0.89$). Both models show significant differences between the scaling exponents for low flows and floods ($\beta_L < \beta_F$ and $\delta_L < \delta_F$), consistent with unregulation in the Tapajos basin.

The use of $A$ as the scale parameter relies on the idea that it represents the horizontal area over which precipitation falls. Using $LA$ is conceptually consistent with this same idea, because $LA$ describes the area through which evapotranspiration is transferred to the atmosphere. $LA$ is an important descriptor of differences between forest and non-forest cover. Our focus on forests is because these ecosystems are highly threatened worldwide (Hansen et al., 2010, 2013; Malhi et al., 2014), while there are important uncertainties about the potential consequences of forest loss on continental water balances (e.g. Bonan, 2008; Ellison et al., 2012; Makarieva et al., 2013; Zhang et al., 2016), including the possibility of forest loss tipping points (Boers et al., 2017; Zemp et al., 2017; Khanna et al., 2017; Lawrence and Vandecar, 2015).

Using $LA$ instead of $A$ as the scale parameter has practical implications for future studies. Using $LA$ allows to explore the influence of a changing scale parameter. $LA$ is much more sensitive to global change than $A$, at time scales that are relevant for decision-making processes. Although studying this sensitivity is out of the scope of our present study, present results provide basis for future studies.

### 4.2 The "Forest Reservoir" hypothesis

The less regulated river basins, Tapajos and Madeira, are also the ones with the less forest cover (Fig. 5a). Forest cover is not an static characteristic of river basins, so different values of the forest cover fraction can be assigned to each basin depending

on the selected data source and time: we use 2003 data from Soares-Filho et al. (2006) —2003 is within the range of all of the studied river flow records—. However, what is important to our argument is not the precise value of the forest cover fraction in each basin, but the observation that, among the Amazon tributaries, the Tapajos and Madeira river basins have experienced large forest cover reductions mainly as a result of forest loss and/or degradation along the so-called arc-of-deforestation in south-southeastern Amazonia (Coe et al., 2013; Asner et al., 2010; Costa and Pires, 2010; Soares-Filho et al., 2006). Using 2002–2014 land water data (GRACE, CSR-v 5.0; Tapley et al. (2004)), and 2002–2014 atmospheric water data (ERA-Interim reanalysis; Balsamo et al. (2015)), we also observed that Tapajos and Madeira are the river basins with the higher long-term average variability of the terrestrial water storages (amplitude of the Liquid Water Equivalent Thickness, LWET, Fig. 5b), and the lower long-term average amount of water stored in the atmosphere (column-integrated precipitable water, Fig. 5c). Taken together, these characteristics are consistent with a river basin with lower capacity to store water within the coupled land-atmosphere system. These observations led us to propose the "forest reservoir" hypothesis that relates the regulation level of the Amazonian river basins with their forest cover.

The physical causes for a river basin to be regulated or unregulated are summarized by its capacity for storing water and controlling its release. Analogously, the capacity of artificial reservoirs to regulate river flows depends on its capacity for storing water and operation rules about how to release it (Magilligan and Nislow, 2005). River basins have natural mechanisms to implement these processes of water handling. These mechanisms depend not only on relatively invariant physical attributes (e.g. geomorphological and geological properties), but also on biophysical processes and characteristics of river basins that can be highly sensitive to global change at policy-relevant time scales, such as forest cover in the Amazon (Malhi et al., 2008; Soares-Filho et al., 2006; Guimberteau et al., 2017). Identifying those factors that are both highly sensitive to global change and strongly influential on runoff production is crucial for predicting the potential effects of global change on river flow regimes. Vegetation cover and vegetation-related processes meet these two conditions in many river basins of the world (Sterling et al., 2013; Coe et al., 2009; Piao et al., 2007), and particularly in the Amazon where the role of forests is so relevant that forest loss could force the system beyond a tipping point (Boers et al., 2017; Khanna et al., 2017; Zemp et al., 2017; Lawrence and Vandecar, 2015; Davidson et al., 2012; Hirota et al., 2011; Sampaio et al., 2007).

Forests can exert strong effects on the store and release of water through a variety of mechanisms. These mechanisms include large evapotranspiration fluxes (Caldararu et al., 2012; von Randow et al., 2012; Da Rocha et al., 2009; Carmona et al., 2016) linked to large precipitation recycling ratios (Van der Ent et al., 2010; Eltahir and Bras, 1994), accumulation and redistribution of soil moisture by root systems (Nadezhdina et al., 2010; Lee et al., 2005; Nepstad et al., 1994), strong capacity for stomatal regulation due to the large cumulative surface area of leaves (Berry et al., 2010; Costa and Foley, 1997), production of biogenic cloud condensation nuclei (Pöschl et al., 2010), below-canopy shading and temperature inversions that restrict direct soil evaporation (Henao et al., Submitted), and the surface drag that is caused by the large height of trees and affects the flow of air over the forests (Khanna et al., 2017).

Collectively, these mechanisms imply that forests have a strong potential to enhance the capacity of river basins for storing water and controlling its release, as well as for producing contrasting and time-variable (e.g. seasonally different) effects on the water balance components. These dual and dynamic effects are key for regulation because it requires opposite effects on

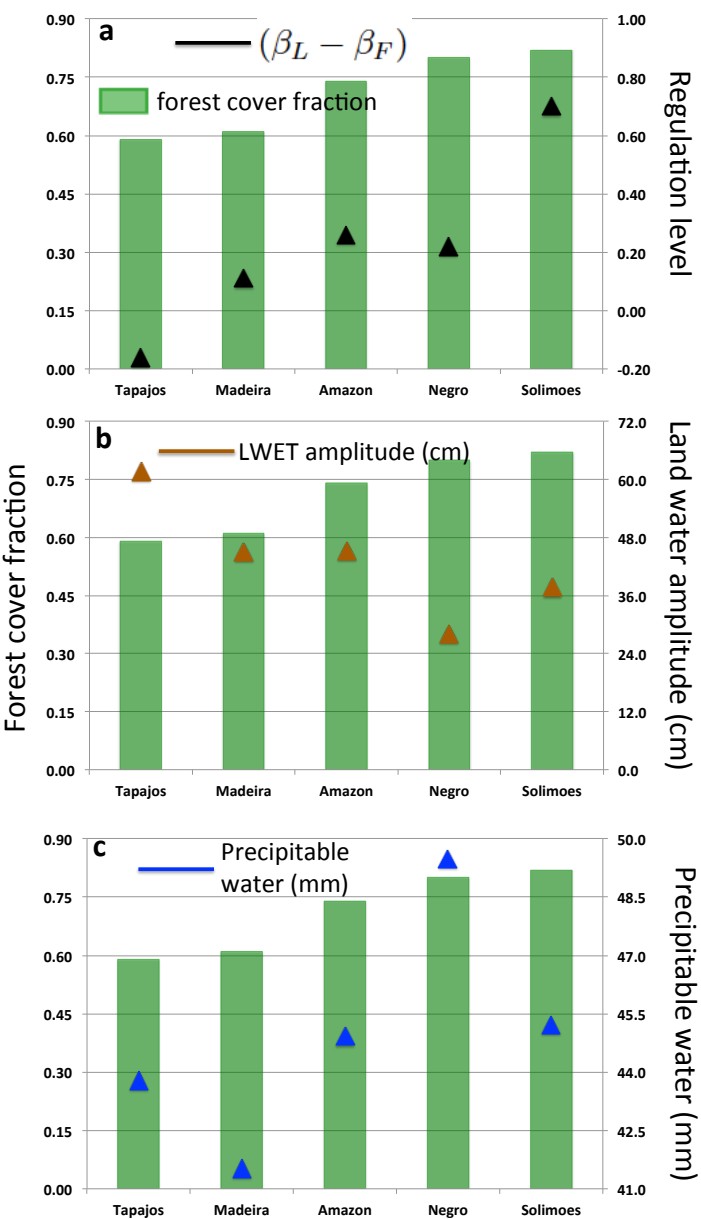

**Figure 5.** Forest cover fraction (2003 data from Soares-Filho et al. (2006)) and (a) the regulation level ($\beta_L - \beta_F$, this study); (b) the long-term (2002–2014) average variability of the land water storages as indicated by the amplitude of the Liquid Water Equivalent Thickness, LWET (data from GRACE, CSR-v 5.0), and (c) the long-term (2002–2014) average amount of atmospheric water as indicated by the column-integrated precipitable water (data from ERA-Interim reanalysis). The Xingu was excluded because the scaling exponents are not significantly different from 1 (Fig 3e).

low flows (amplification) and floods (dampening). The forest reservoir describes *the natural capacity of river basins (in the Amazon or similar basins) to store water and control its release through land-atmosphere interactions (mainly precipitation recycling) that depend strongly on the presence of forests*. This hypothesis considers a river basin as the coupled land-atmosphere system comprising not only the terrestrial fluxes and storages of water but also the atmospheric ones (Fig. 6). Although the capacity of the atmosphere to store water is relatively small, its capacity to transport water within or outside a system is huge (Trenberth et al., 2007). Indeed, in the long term, all continental water comes from the ocean through the atmosphere because the atmospheric fluxes of water are the only ones that flow upstream in river networks, while terrestrial fluxes are directed into the ocean by gravitational forces.

The water balance equation for the forest reservoir control volume (Fig. 6),

$$\frac{d(S_l + S_a)}{dt} = \nabla Q - R, \tag{8}$$

establishes that changes in water storage —including both land ($S_l$) and atmospheric ($S_a$) components— are governed by differences between the net atmospheric moisture convergence ($\nabla Q$, the only input flux) and runoff ($R$, including both surface and sub-surface fluxes, the only output flux). $P$ (precipitation), $E$ (evapotranspiration) and $I$ (infiltration) are not external fluxes but components of complex land-atmosphere interactions (e.g. precipitation recycling) that occur within the system and, therefore, are fundamental to the mechanisms that can explain the capacity of a basin system for regulating river flows. Although external forcings (e.g. climate change or variability effects) do affect the response of the system ($R$ is not independent of $\nabla Q$), the capacity for regulating river flows can only be a consequence of the system's internal dynamics. Otherwise, if the response of a system simply follows external forcings (if $R$ were entirely governed by $\nabla Q$), then there would be no capacity for regulation. Variations in the internal dynamics of water storage allow for the occurrence of different river flow regimes under the same external forcings.

The occurrence of floods or low flows is related, respectively, to the abundance or scarcity of water, which depend on external forcings that determine whether $\nabla Q$ is large or small during any given time period (e.g. wet and dry seasons). Floods dampening depends on the capacity of the basin to retain water when $\nabla Q$ is large (wet season), which implies increasing water storage, consistent with

$$\frac{d(S_l + S_a)}{dt} \begin{cases} > 0, & \text{if } \nabla Q > R \text{ (floods dampening via water storage)} \\ \leq 0, & \text{if } \nabla Q \leq R \text{ (no dampening or even amplification of floods).} \end{cases} \tag{9}$$

Analogously, low flows amplification depends on the basin's capacity for releasing previously-stored water when $\nabla Q$ is small (dry season), therefore reducing water storage as described by

$$\frac{d(S_l + S_a)}{dt} \begin{cases} \geq 0, & \text{if } \nabla Q \geq R \text{ (no amplification or even dampening of low flows),} \\ < 0, & \text{if } \nabla Q < R \text{ (low flows amplification via water release).} \end{cases} \tag{10}$$

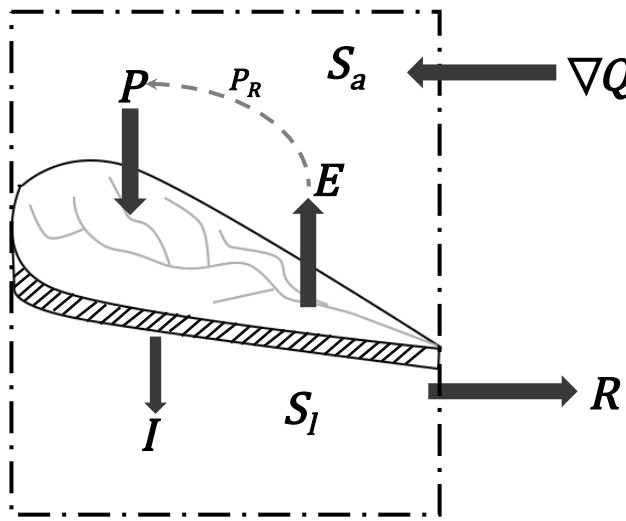

**Figure 6.** Forest reservoir control volume including the coupled land-atmosphere basin system. The system exchanges water with its exterior through the net atmospheric moisture convergence ($\nabla Q$) and runoff ($R$ which includes surface and sub-surface fluxes). $P$ (precipitation), $E$ (evapotranspiration) and $I$ (infiltration) are internal fluxes that determine the distribution of water storage between land ($S_l$) and atmospheric components ($S_a$). Precipitation recycling ($P_R$) can occur within the system.

The importance of forests for the system's internal dynamics of water storage is highlighted by their relation with precipitation. Precipitation is not entirely determined by external forcings nor independent of the presence of forests. If precipitation regimes were independent of forest-related processes, then those regimes should not significantly change in response to forest cover change. This is contradicted by an increasing body of scientific evidence indicating that forest cover change can

significantly alter precipitation regimes in the Amazon (Zemp et al., 2017; Lawrence and Vandecar, 2015; Spracklen and Garcia-Carreras, 2015; Lima et al., 2014; Makarieva et al., 2013; Stickler et al., 2013; Costa and Pires, 2010; Coe et al., 2009; Makarieva and Gorshkov, 2007). Through its impact on precipitation, forest cover change can affect all other water balance fluxes (e.g. river flows; Lima et al., 2014; Stickler et al., 2013; Coe et al., 2009), as well as terrestrial and atmospheric storages. Notably, the simulated impacts of deforestation on river flows can be opposite depending on whether the precipitation response

to deforestation is included or not (Lima et al., 2014; Coe et al., 2009).

Recycled precipitation ($P_R$) is a key factor for regulation because it represents a potentially large amount of water that can be retained within the system through land-atmosphere circulation (Fig. 6). Therefore, in largely forested basins, the precipitation recycling ratio is indicative of the importance for regulation of the forest-mediated land-atmosphere interactions. Global estimates indicate that land evaporation accounts for about half of continental precipitation (Gimeno et al., 2012; Van der

Ent et al., 2010), whereby forests are major contributors (Schlesinger and Jasechko, 2014; Bonan, 2008). In the Amazon river basin, recycled precipitation also accounts for about half of the total precipitation (Eltahir and Bras, 1994). With this amount of forest-related precipitation, a disruption of the recycling mechanism has a strong potential to modify the internal dynamics of water transport and storage that control river flows regulation (e.g. Zemp et al., 2017).

Precipitation recycling is not a dominant process at all spatial and temporal scales in every basin of the world. It is difficult to quantify the degree to which terrestrial evapotranspiration supports the occurrence of precipitation within a certain region, partly because this mechanism has characteristic time and length scales, and depends on the size, shape and location of basins, as well as on the atmospheric pathways of moisture transport (Van der Ent and Savenije, 2011). However, it is
widely-recognized that precipitation recycling is a crucial process in the hydrological cycle of the Amazon and neighbouring basins (Martinez and Dominguez, 2014; Zemp et al., 2014; Eltahir and Bras, 1994). All of the studied large basins are sinks (receive recycled precipitation) and sources (feed recycled precipitation through evapotranspiration) of significant amounts of continental moisture, with impacts that can be spread throughout the continent by complex cascading effects that are sensitive to forest cover change (Zemp et al., 2017, 2014). Global estimates indicate the length scale of precipitation recycling can be as
low as 500 km in tropical regions (Van der Ent and Savenije, 2011), which is not excessively large compared with the size of the basins. The observed seasonal variability of atmospheric moisture pathways over South America allows for the occurrence of significant precipitation recycling all over the Amazon basin (Zemp et al., 2014; Arraut et al., 2012).

Our conclusion that the Madeira and Tapajos are the less regulated basins, with Tapajos being unregulated (Table 1), relies only on the observed values of the scaling exponents, following the theoretical framework developed in Section 2. Therefore,
this conclusion does not ignore the important role of geological and geomorphological processes (Miguez-Macho and Fan, 2012; Bruijnzeel, 2004). Depending on the case study, different levels of regulation or transitions between states could be attributed to different causes. The forest reservoir hypothesis provides a potential explanation linking forest cover and river flow regulation. The idea is not that the effect of land cover (particularly forest cover in the Amazon) on river flows regulation is stronger than any other effect (e.g. geological and geomorphological effects), but that the role of land cover is not negligible
and critically important because of its sensitivity to global change, especially in a region such as the Amazon where forest ecosystems are highly threatened and forest-related precipitation recycling plays a major role (Davidson et al., 2012). We foresee a potential danger in the assumption that the regulation capacity of river basins depends on geomorphological and geological processes with land cover playing a negligible role. Under this assumption, land cover change (e.g. forest loss) would not change the capacity of river basins to regulate river flows.

The forest reservoir mechanisms may have been previously overlooked because the size of the atmospheric storage is much smaller than that of the terrestrial storage ($S_a << S_l$; Trenberth et al. (2007)), and also because the size of the terrestrial storage (e.g. aquifer systems) is mainly determined by geological and geomorphological properties. However, the key factor for regulation is not the size of the atmospheric storage but the possibility of retaining large amounts of water within the system through land-atmosphere interactions.

**4.3  Forest loss effects on regulation: a potential critical threshold**

Forest loss does not reduce or increase river flows in every basin at every temporal and spatial scale (Zhang et al., 2016; Ellison et al., 2012; Zhou et al., 2015). Fundamental reasons for this are that forests have an inherent capacity to either increase or decrease the water balance components, and that these effects have a complex and dynamic nature. For instance, forests can increase or decrease $E$ via, respectively, opening or closing stomata, which is related to water availability: stomatal aperture

tends to be increased during drought stress and decreased during excessive water stress (Cornic, 2000; Lambers et al., 2008). Further, forest loss can significantly alter the hydraulic properties of soils, especially by reducing infiltrability (Zimmermann et al., 2006). Through these impacts, forest loss can alter all the water balance components in complex ways. If the effect of forest loss were always to reduce $E$ (due to reduction of the cumulative leaf area) with no impact on $P$ (as implicitly assumed in hydrological models that use $P$ as a fixed input) nor on the hydraulic properties of soils and regulation capacity of the basin, then forest loss should be always associated with increased $R$ and, therefore, increased floods and low flows. Likewise, if the effect of forest loss were always to increase $E$ (related to, e.g. weaker stomatal regulation, disruption of below canopy shading and stability, and increased wind speed over the surface) with no other effects, then forest loss should always lead to reduced $R$ and, therefore, reduced floods and low flows. In both cases, the effect of forest loss on extreme river flows would always be in the same direction. In contrast, the forest reservoir hypothesis considers that forest loss can have contrasting effects on low flows and floods, mainly because the production of these extreme flows is governed by different processes occurring during different seasons.

The forest reservoir hypothesis implies that the regulation capacity of a river basin can be importantly sensitive to forest cover change. The size of artificial reservoirs determines their regulatory capacity. Likewise, the regulatory capacity of the forest reservoir depends on its size, which is related to the extent of forest cover. This implies that forest loss weakens regulation. The lower levels of regulation in the Madeira and Tapajos river basins (Table 1) are consistent with a weaker forest reservoir (these two basins are the less forested ones, Fig. 5a), likely related to extensive forest loss that has occurred along the arc-of-deforestation (Coe et al., 2013; Asner et al., 2010; Costa and Pires, 2010; Soares-Filho et al., 2006). Notably, these less regulated basins are also the ones with more large artificial reservoirs in operation (http://dams-info.org/). The introduction of artificial reservoirs can cause contrasting effects on regulation. Assuming that an artificial reservoir is operated so as to reduce floods and increase low flows, its introduction in a river basin should enhance river flow regulation. However, the construction of reservoirs is usually linked to other human activities —e.g. road construction, and associated agricultural expansion and deforestation (Soares-Filho et al., 2006; Mahe et al., 2005)— that can reduce the natural capacity of river basins to regulate river flows. Our results suggest that this is the case in the Madeira and Tapajos basins.

Forest loss does not weaken regulation because it changes the capacity of the atmospheric and terrestrial water storages, but mainly because it reduces the capacity of the basin system (Fig. 6) to retain water through its complex internal dynamics of land-atmosphere interactions. Figure 7 shows a conceptual example of how forest loss can disrupt river flow regulation (increase the extremes amplitude) via weakening the forest reservoir. Forest loss can exacerbate floods by increasing $R$ through reduction of $E$ and $I$ during the wet season when $P$ is large due to large $\nabla Q$ (Fig. 6 and Fig. 7a). $E$ and $I$ reduction can be associated, respectively, with reduced leaf area and infiltrability. $E$ reduction can weaken $P$ recycling as a mechanism for dampening floods by recirculating water within the system. These effects are consistent with an enhanced conversion of $P$ into $R$ during the wet season and, therefore, enhanced floods and reduced water storage. This is described by Eq. (9) where floods are not dampened if water storage ($S_l + S_a$) is not increased. Water storage reduction during the wet season results in a decreased capacity of the system to amplify low flows via base flow during the dry season (Fig. 7b). Amplifying low flows when $\nabla Q$ is relatively small (the dry season) requires the release of water that has been previously stored, consistent with

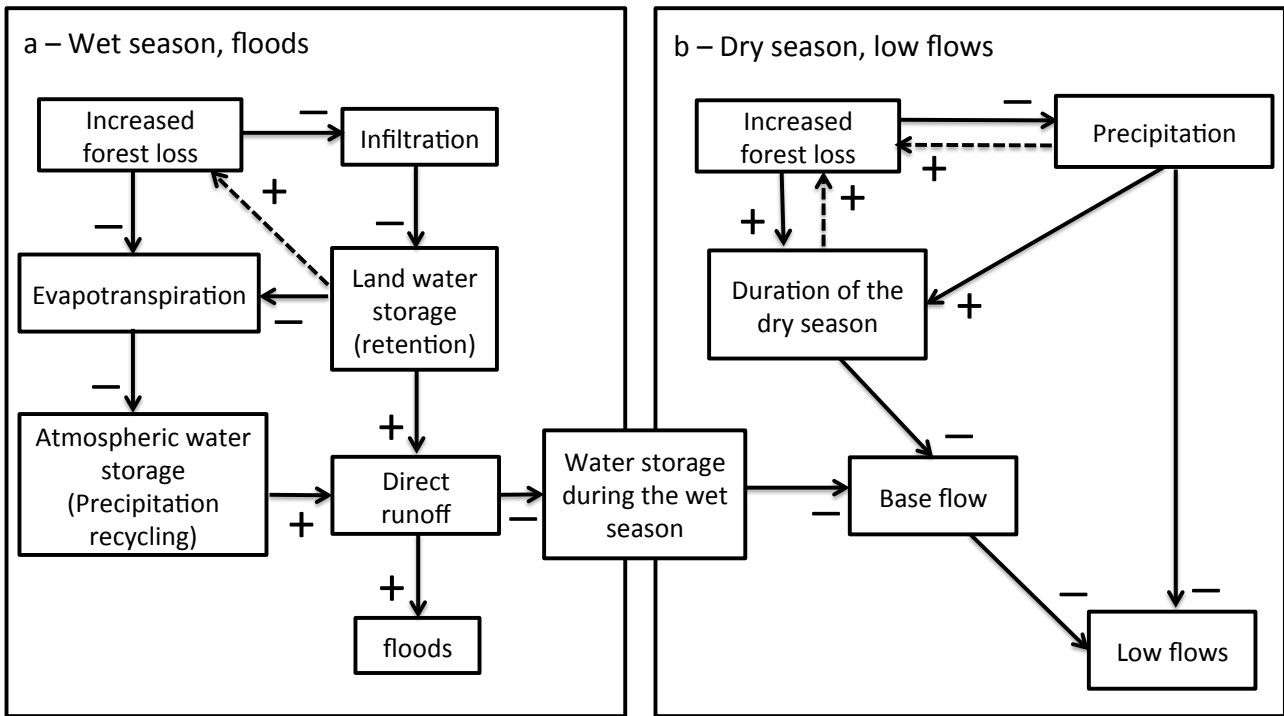

**Figure 7.** Potential weakening of the forest reservoir due to forest loss. (a) The loss of forests can exacerbate floods through increases of the direct runoff associated with reductions of the evapotranspiration and infiltration fluxes. These effects are associated with a reduction of the capacity of the coupled land-atmosphere system for retaining water during the wet season. (b) Less water retention during the wet season can reduce the base flow during the dry season. The loss of forests can reduce low flows through reductions of base flow and precipitation, both of them associated with a reduction of the capacity of the coupled land-atmosphere system for storing and releasing water during different periods of time. Dashed arrows indicate potential positive feedbacks to forest loss.

$d(S_l + S_a)/dt < 0$ in Eq. (10). Deforestation-induced reduction of $P$ (Spracklen and Garcia-Carreras, 2015) or lengthening of the dry season (Lima et al., 2014; Costa and Pires, 2010), consistent with a disruption of the wet season onset (Wright et al., 2017), can further reduce low flows.

The forest reservoir hypothesis implies that forest loss can increase floods while reducing low flows (Fig. 7). This is not
5  inconsistent with increasing scientific evidence that large-scale forest loss will reduce $P$ over the Amazon (Spracklen and Garcia-Carreras, 2015). Reduced $P$ can explain a decrease of low flows but does not necessarily imply a decrease of floods too. Floods depend importantly not only on the total amount of $P$ but also on its temporal distribution (rainfall intensity and duration) and the hydraulic properties of the surface (Reed, 2002). Variations in the capacity of the basin system for retaining and releasing water during wet and dry seasons allow for the occurrence of larger floods with smaller $P$. A comparable
10  situation has been observed in the Nakambe River in Africa where reduced precipitation has lead to the counter-intuitive effect of increased floods, even despite an increase in the number of dams in the river basin (Mahe et al., 2005).

The identification of alternative regulation states from scaling properties in river basins (Section 2), together with the hypothesis that forest loss weakens the regulatory capacity, imply that forest loss can cause a transition from the regulated state to the unregulated state. This also implies that there is a forest cover critical threshold where the transition occurs. In our results, the forest cover fraction in the less regulated basins is $\sim 0.60$, while in the more regulated basins it is $> 0.70$ (Fig. 5a), which suggests a possible range for the critical threshold. Although more-detailed studies are essential to understand regulation dynamics in different regions, as well as to identify potential critical thresholds, our analysis shows that scaling patterns may be used to characterize regulation states and infer transitions in river basins. Such empirical approaches are essential (e.g. Hirota et al., 2011) because it is becoming clear that accurate mechanistic models to predict critical thresholds (or tipping points) are currently beyond our reach (Scheffer et al., 2009), and the detection of early-warning signals for critical transitions in complex systems (e.g. river basins) remains a fundamental challenge in environmental science today (Scheffer et al., 2009; Lenton, 2011).

## 5 Conclusions

We have shown how the scaling properties of mean and extreme river flows are a signature of the river flow regime in any river basin. Through the values of the scaling exponents, a river basin can be classified as regulated or unregulated, depending on whether it dampens or amplifies extreme river flows, respectively. These scaling exponents are sensitive to global change, so a river basin can shift from the regulated to the unregulated state. The scaling exponents provide a metric for the proximity to the critical threshold. Our results indicate that environmental perturbations that reduce the natural capacity of river basins to regulate river flows, tend to increase the scaling exponent for floods and to decrease that for low flows. This provides a prediction of the direction of change in the scaling exponents of river basins as a result of global change, that can be used to design and simulate scenarios of future river flow regimes. The theoretical basis of our physical interpretation of the scaling properties of river flows is generally applicable to any river basin.

We have applied the proposed interpretation of river flows scaling properties to the Amazon river basin and found both the regulated (all except the Tapajos) and unregulated (the Tapajos) states among its main tributaries. Then we proposed the forest reservoir hypothesis to describe the natural capacity of river basins to regulate river flows through land-atmosphere interactions (mainly precipitation recycling) that depend strongly on the presence of forests, especially in the Amazon. A critical implication of this hypothesis is that forest loss can force the Amazonian river basins from regulated to unregulated states. This provides further evidence about the possible outcome of widespread forest loss in the Amazon, potentially involving forest loss critical thresholds, a matter of great uncertainty and concern (Boers et al., 2017; Khanna et al., 2017; Zemp et al., 2017; Lawrence and Vandecar, 2015; Davidson et al., 2012; Hirota et al., 2011).

These results provide foundations and a quantitative basis for using the scaling theory in solving four fundamental challenges in river basin science: the "PUB problem" that extends to every river basin in a changing environment (Hrachowitz et al., 2013; Gupta et al., 2007); the detection of early-warning signals of critical thresholds in river basins (Lenton, 2011; Scheffer et al., 2009); the production of parsimonious river basin classifications based on dimensionless similarity indices —the scaling

exponents— or dominant processes —amplification or dampening of extreme river flows— (McDonnell et al., 2007); and the exploration of the organizing principles that underlie the heterogeneity and complexity of river flow production processes in river basins with different hydroclimatic regimes, and at different scales (Blöschl et al., 2007; McDonnell et al., 2007). We addressed this by advancing from observed patterns (Figs. 2–5) to processes: the forest reservoir hypothesis (Figs. 6,7), as

recommended by Sivapalan (2005).

*Data availability.*  All data for this paper is properly cited and referred to in the reference list.

*Author contributions.*  JFS, JCV, AMR and GP designed the research. JFS and AMR developed the mathematical model. JFS, ER, IH and DM performed data analysis. JFS developed the forest reservoir hypothesis and wrote the manuscript with input from other authors. All authors discussed the results and conclusions.

*Competing interests.*  The authors declare that they have no conflict of interest.

*Acknowledgements.*  We gratefully acknowledge constructive comments from Editor Patricia Saco and two anonymous referees. Funding was provided by "Programa de investigación en la gestión de riesgo asociado con cambio climático y ambiental en cuencas hidrográficas" (UT-GRA), Convocatoria 543-2011 Colciencias. JFS was partially supported by the IAI-INPE Internship program: "Understanding Climate Change and Variability in the Americas". AMR was partially supported by Colciencias grant 115-660-44588. JCV was partially supported

by NSF- EF-1340624 through the University of Arizona.

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
