# Peer review of "Scaling properties reveal regulation of river flows in the Amazon through a "forest reservoir""

_Hydrology and Earth System Sciences, 2017_

## Referee Comment (RC1) · Anonymous Referee #1 · 3 Jul 2017

The authors propose a method to classify large rivers of Amazonia according to their capacity to attenuate low, mean and high flows. Then, they introduce the hypothesis of "forest reservoir" with the premise that land use and land cover changes can induce changes of the river regimes from regulated to unregulated states. The idea is interesting, and can provide new insights on the sensibility of river to land use changes, besides other well-known indexes such as river elasticity. However, I have various concerns about the manuscript and the proposed index, which need to be clarified and analyzed:

1) Regarding the idea of classification of large rivers based on a scaling property defined as the product of LAI and Drainage area, it should be noted that traditional approaches used in hydrology (for instance, Molinier et al. 1996 for the Amazon Basin)

have shown that there are well-known relationships between discharge and drainage area. These relationships stands across scales simply because they are based on the continuity equation. Perhaps the new approach in the manuscript is to compare of regional indexes for low, mean and high flows. Having say so, in my opinion, there is no need to consider the LAI in the scaling parameters, once the drainage area explain most of the variability. In mathematical terms, LAI works like a constant value with no effect on the relationships, because LAI is fixed in each basin

2) In the paper of Molinier et al (1996), regional differences among the response between Amazon tributaries are explained in terms of the rainfall regime. Given the large size of the Amazon, the annual distribution of rainfall varies from northern to southern tributaries. In this context, it is not a surprise that the southern tributaries are generally less regulated than the northern tributaries, considering that the annual distribution of rainfall is relatively regular in northern Amazonia (due to the effect of the ITCZ), while in southern and eastern Amazonia seasonal variability is higher (related to the South American monsoon circulation).

3) Besides the effects of rainfall regimes across the Amazon, the Basin can be divided in contrasting morpho-structural units (Molinier et al 1996). The implication of this is relevant in terms of the ability of the basin to accumulate water in the groundwater system and, consequently, in the capacity of Amazon tributaries for attenuating floods and droughts. This has been explore in detail by Miguez-Macho and Fan (2012), who showed that headwater responses are dominated by the effect of groundwater, while on large floodplains there are a two-way fluxes between surficial water and groundwater. This is why in several areas, such along the Solimoes and Amazonas, the degree of regulation is higher: those areas are characterized by large floodplains in combination with powerful aquifer systems (the Solimoes and Alter-do-Chao Formations). In other words, damping effects on the river main stem are the result of geological and geo-morphological processes.

4) Finally, the authors suggest a series of change due to the effect of land use and

land cover changes on the ability of rivers to regulate the response. Although this is a conceptually sound hypothesis, demonstration based on river data observations, rather than modeling, have proven to be quite illusive in large basins. For instance, detailed analysis of trends on rivers (Marengo, 2009; Espinoza et al. 2009) showed that the trends detected are associated to interdecadal variability, instead of the potential effects of land use changes. Moreover, recent trends in the hydrological cycle of the Amazon have also been attributed to the warming of the Atlantic Ocean (Gloor et al. 2015) rather than local-scale changes. If we take into consideration that most of the ability to regulate river regimes is related to rainfall regimes and geological – geomorphological characteristics ,as demonstrated before, it might be challenging to disentangle LUCC effects from those major natural drivers.

In conclusion, it is my opinion that the manuscript should go through a major revision. The authors need to explain better the role of LAI in the relationships they proposed (which I think is unnecessary). Regarding the forest reservoir hypothesis, the concept should be clearer if an example based on observations is brought into the manuscript. I presume that, along the "deforestation arc", several candidate basins undergoing through severe land use and land cover changes in the last decades can be found.

References

Espinoza Villar JC, Guyot JL, Ronchail J, Cochonneau G, Filizola N, Fraizy P, Labat D, Oliveira E, Ordoñez JJ, Vauchel P. 2009. Contrasting regional discharger evolutions in the Amazon basin (1974–2004). Journal of Hydrology 375: 297–311. DOI: 10.1016/j.jhydrol.2009.03.004.

Gloor, M., J. Barichivich, G. Ziv, R. Brienen, J. Schöngart, P. Peylin, B. B. Ladvocat Cintra, T. Feldpausch, O. Phillips, and J. Baker (2015), Recent Amazon climate as background for possible ongoing and future changes of Amazon humid forests, Global Biogeochem. Cycles, 29, 1384–1399, doi:10.1002/2014GB005080.

Marengo JA. 2009. Long-term trends and cycles in the hydrometeorology of the Amazon basin since the late 1920s. Hydrological Processes 23(22): 3236–3244. DOI: 10.1002/hyp.7396.

Miguez-Macho, G., and Y. Fan (2012), The role of groundwater in the Amazon water cycle: 1. Influence on seasonal streamflow, flooding and wetlands, J. Geophys. Res., 117, D15113, doi:10.1029/2012JD017539.

Molinier M, Guyot JL, Oliveira E, Guimarâes V. 1996. Les regimes hydrologiques de l'Amazone et de ses affluents. In L'hydrologie tropicale: g'eoscience et outil pour le d'eveloppement. IAHS Publishers: Mai, Paris; 209–222.

---

## Referee Comment (RC2) · Anonymous Referee #2 · 21 Jul 2017

This work by Salazar et al. is an interesting study and presents original ideas. The authors study scaling properties of river flows of the Amazon basin and its subbasins. Identifying whether a basin attenuates or amplifies extremes in the flow regime, they propose that (Amazonian) river basins can go through tipping points of river flow regulation if forest loss exceeds a critical level.

Despite the interesting features of this study, I have a number of concerns that leave me yet unconvinced of some of the interpretations and conclusions drawn from them. These issues need to be addressed in a major revision before I can recommend publication of this manuscript.

The authors hypothesize that, generally, river flows in Amazonian basins are regulated by the forests, meaning that extreme lows and highs in flows are attenuated by the forest. Mainly forest-induced precipitation recycling would be responsible for this attenuation. Indeed, this provides a positive feedback between the land/biosphere and rainfall, and such positive feedbacks are necessary for tipping points to occur (Van Nes et al. 2016, Trends in Ecology & Evolution 31:902-904). It should, however, be made more clear how land-atmosphere interactions cause both higher minima and lower maxima in river flows. Moisture recycling also has a typical spatial scale and direction. How does this affect the regulation of river flows and could the finding that the Tapajos is unregulated be an artefact of its size (and possibly shape)?

Key for understanding the feedbacks in the system should be figure 6. However, it rather confused me, for the signs of the arrows do not seem to represent the sign of individual interactions: for example, evapotranspiration does not decrease (as is indicated now), but increase atmospheric water storage. And how could atmospheric water storage increase direct runoff? Both the figure and the text should be revised to guide the reader more to understand the core of the idea that is proposed.

Furthermore, I am not convinced of the threshold of 60% tree cover below which river basins shift from regulated to unregulated. The evidence for this threshold is that the Tapajos, inferred to be the only unregulated subbasin in the Amazon, is also the only one with an average tree cover of below 60%. This correlation is too weak to draw the conclusion that this threshold exists, let alone that deforestation has caused the Tapajos to pass a tipping point.

The authors also relate the 60% threshold to its correspondence to the threshold that can separate forests and savannas as alternative stable states. However, the latter threshold applies at local scales instead of at basin scale. A basin-scale average tree cover does not provide information about how far from such a threshold a forest is in any particular location; having a larger extent of grasslands in a basin does not necessarily mean that the forests in the basin are closer to a threshold. Indeed, the southern subbasins have more naturally occurring savannas and therefore lower average (subbasin-sclae) tree cover. The presence of these savannas is a result of rainfall

seasonality (Staver et al. 2011, Science 334:230-232), which, as pointed out by referee 1, itself affects the regulation of river flows.

If the hypothesis that river flow regulation can pass tipping points holds, what would be the concrete consequences of such transitions? Obviously the limit case of infinitely high and low river flows will not be reached, so how do the authors see the future of the Tapajos and other basins if land use change continues? The paper lacks explicitness in this sense, which will leave readers like myself to question the validity of the forest reservoir concept.

Minor points:

In figure 5 the bar charts for tree cover are presented relative to a baseline of 60%, suggesting independent evidence for such a baseline, whereas the results in the figure itself are the evidence for a threshold of 60%. Please change to bar charts for tree cover starting at 0.

In figure 3e, the dots indicate that the exponents are significantly different. Yet, it is also said that it cannot be rejected that the exponents differ from 1. One of these statements must be wrong.

---

## Author Comment (AC1) · 27 Jul 2017

Comment:

The authors propose a method to classify large rivers of Amazonia according to their capacity to attenuate low, mean and high flows. Then, they introduce the hypothesis of "forest reservoir" with the premise that land use and land cover changes can induce changes of the river regimes from regulated to unregulated states. The idea is interesting, and can provide new insights on the sensibility of river to land use changes, besides other well-known indexes such as river elasticity. However, I have various concerns about the manuscript and the proposed index, which need to be clarified and analyzed:

[Figure]

Response:

We thank the reviewer for his/her comments. Specific answers to each comment are provided below.
* * *
Comment:

1) Regarding the idea of classification of large rivers based on a scaling property defined as the product of LAI and Drainage area, it should be noted that traditional approaches used in hydrology (for instance, Molinier et al. 1996 for the Amazon Basin) have shown that there are well-known relationships between discharge and drainage area. These relationships stands across scales simply because they are based on the continuity equation. Perhaps the new approach in the manuscript is to compare of regional indexes for low, mean and high flows. Having say so, in my opinion, there is no need to consider the LAI in the scaling parameters, once the drainage area explain most of the variability. In mathematical terms, LAI works like a constant value with no effect on the relationships, because LAI is fixed in each basin

Response:

We agree that the scaling relations between drainage area and river flows are well-known. Indeed, this is acknowledged in the Introduction (starting at line 17 of p.1). The main question that we are addressing is not whether the scaling relations are valid in the Amazon (this was expected), but how to interpret the values of scaling parameters in terms of physical processes. Despite important advances (e.g. Gupta and Dawdy, 1995), this remains an open scientific question, with important practical implications related to the prediction of hydrological consequences of global change.

The first aim of our paper is to provide a physical interpretation of the scaling properties (particularly the scaling exponents) that is novel (most previous studies have focused on the interpretation of the scaling exponents for floods only), and widely-applicable

to different basins worldwide (the only assumption is that river flows in a given river basin exhibit scaling properties through power laws of the form of equation (1)). This interpretation (presented in Section 2) does not require the use of the cumulative leaf area, LA, as the scale parameter. Instead, it allows to investigate the use of different scale parameters. This is why all of the equations in Section 2 (equations 1 through 6) use S as a general scale parameter that could be replaced, for instance, by A (the drainage area), LA, or other, depending on the case study.

Our general idea about the classification of river basins is not based on the use of LA as the scale parameter (LA is not used in Section 2). Scaling properties, represented by the values of the scaling exponents for mean and extreme river flows, provide a parsimonious description of river flow regime of any river basin and allow its classification as regulated or unregulated. LA was used as the scale parameter for the application of our general framework (presented in Section 2) to the particular case of the Amazon (presented in Section 3). However, we verified the consistency of our results when using A instead of LA as the scale parameter (Supplementary Figures S1 through S6 show the scaling relations using both A and LA for each basin). Our idea is not that LA should be used as the main scale parameter in any river basin, but that it can be successfully used as a scale parameter in the Amazon.

We consider that leaf area is a key biophysical attribute with a strong influence on the hydrological cycle in the Amazon. Evidence of this statement is that most climate models use leaf area as a parameter to describe the hydroclimatic consequences of land cover change in the Amazon. As stated in the paper: "Large scale forest degradation or loss is a major driver of environmental change in these river basins [References in the paper]. The capacity to maintain high evapotranspiration rates is a key attribute of Amazonian forests associated with their large cumulative area of leaves [References in the paper]. We take this into account by setting the [scale] parameter as S=LA..." (lines 9-13, p. 4).

Using LA instead of A as a scale parameter has practical implications for future stud-

ies. Using LA allows to explore the influence of a changing scale parameter. LA is much more sensitive to global change than A, in temporal scales that are relevant for decision-making. LA varies with time as a consequence of rapid ongoing global-change-related processes such as deforestation, forest die-off and forest degradation. Although studying this variability is out of the scope of our present study, we consider that our reported results provide basis for future studies.
* * *
Comment:

2) In the paper of Molinier et al (1996), regional differences among the response between Amazon tributaries are explained in terms of the rainfall regime. Given the large size of the Amazon, the annual distribution of rainfall varies from northern to southern tributaries. In this context, it is not a surprise that the southern tributaries are generally less regulated than the northern tributaries, considering that the annual distribution of rainfall is relatively regular in northern Amazonia (due to the effect of the ITCZ), while in southern and eastern Amazonia seasonal variability is higher (related to the South American monsoon circulation).

Response:

We appreciate that the reviewer agrees with our result that Tapajos and Madeira (the southern tributaries) are the less regulated basins. This implies that our interpretation of the scaling properties (Section 2) of the Amazon and its main tributaries (Section 3) worked well on distinguishing the more and less regulated basins.

The precise meaning of the comment about the seasonal variability of rainfall is not entirely clear to us. The intra-annual variability of rainfall is very pronounced all across the Amazon (e.g. Fig. 5 of Espinoza et al., 2009; and Fig. 2 of Molinier et al., 1996). We have assumed that the reviewer refers to the amplitude of the annual cycle of precipitation, which tends to be higher in the south. More or less amplitude of the rainfall seasonal cycle do not necessarily imply greater or lower capacity of a basin to regulate

river flows. For instance, Tapajos and Madeira are both located in the south, but their scaling properties indicate that they are significantly different in terms of regulation. While Madeira is regulated (mainly due to its capacity to dampen floods), Tapajos is unregulated (mainly because it dampens low flows). If the capacity of the Tapajos basin system to store water and control its release were big enough, then the basin would behave as regulated despite its seasonal rainfall regime. Our idea is that such capacity is importantly (not exclusively) dependent on the biophysical processes related to land cover, especially forest cover. Notably, the most regulated basin (Solimoes) is not a northern tributary, its drainage area ranges from southern to northern Amazonia. Further, the northernmost basin (Negro), is not the most regulated either. In our response to the fourth comment we include a conceptual example considering whether the regulation capacity of a basin could change due to forest loss, even without changing the rainfall regime.

The capacity of a system to regulate its response must be an internal property of the system rather than a consequence of external forcings. Seasonality is essentially a result of external forcings related to climatic effects of solar radiation variations. The capacity to regulate implies the capacity of a system to modify its response via its internal dynamics. If the response simply follows external forcings, then there is no capacity for regulation. The system that we are considering in our hypothesis of the forest reservoir concept "considers a river basin as the coupled land-atmosphere system comprising not only the terrestrial fluxes and storages of water but also the atmospheric ones" (lines 29-31, p. 10). Therefore, precipitation is not an external flux but an internal one, and the land-atmosphere interactions (e.g. those involved in precipitation recycling) occur within the system. Our idea is that these interactions are part of the mechanisms that explain the capacity of the system to store water and control its release (release outside the system through river flows), i.e. the capacity for regulating river flows.
* * *
Comment:

3) Besides the effects of rainfall regimes across the Amazon, the Basin can be divided in contrasting morpho-structural units (Molinier et al 1996). The implication of this is relevant in terms of the ability of the basin to accumulate water in the groundwater system and, consequently, in the capacity of Amazon tributaries for attenuating floods and droughts. This has been explore in detail by Miguez-Macho and Fan (2012), who showed that headwater responses are dominated by the effect of groundwater,while on large floodplains there are a two-way fluxes between surficial water and groundwater. This is why in several areas, such along the Solimoes and Amazonas, the degree of regulation is higher: those areas are characterized by large floodplains in combination with powerful aquifer systems (the Solimoes and Alter-do-Chao Formations). In other words, damping effects on the river main stem are the result of geological and geomorphological processes.

Response:

Following on our response to the first comment, the first aim of our paper is not to explain the causes of river flow regulation but to show how river flow regulation can be assessed from scaling properties (Section 2). This interpretation of the scaling properties does not ignore the role of geological and geomorphological processes. The scaling properties are based on river flow observations that are the result of all of the biophysical processes playing a role in the hydrological cycle. Therefore, depending on the case study, different levels of regulation (identified through scaling properties) might be attributed to different causes. As stated in the paper: "The physical causes for a river basin to be regulated or unregulated are summarized by its capacity for storing water and controlling its release. [...] River basins have natural mechanisms to implement these processes of water handling. These mechanisms depend not only on relatively invariant physical attributes (e.g. geomorphological and geological properties), but also on biophysical processes and characteristics of river basins that can be highly sensitive to global change at policy-relevant time scales, such as forest cover in the Amazon [References in the paper]." (lines 16-21, p.10).

We agree that geomorphological and geological processes affect the capacity of basins to regulate river flows. However, we also propose that the role of forests (particularly in the Amazon) in determining the regulation capacity of a river basin cannot be neglected. This discussion has important practical implications. As stated in the paper: "Identifying those factors that are both highly sensitive to global change and strongly influential on runoff production is crucial for predicting the potential effects of global change on river flow regimes. Vegetation cover and vegetation-related processes meet these two conditions in many river basins of the world [References in the paper], and particularly in the Amazon where the role of forests is so relevant that forest loss could force the system beyond a tipping point [References in the paper]". From this perspective, the regulation capacity of a river basin may be more sensitive to land cover change-related processes. In synthesis, the regulation capacity of a river basin could be mostly dependent on land cover-independent geological and geomorphological attributes (relatively invariant), while importantly sensitive to land cover (highly dynamic).

We consider that the paper by Miguez-Macho and Fan (2012), a very good modelling study, does not allow to produce a "definite" explanation about how and why the regulation capacity of river basins in the Amazon differs between one another. This is mainly because of the limitations of the coarse modelling study that the same authors acknowledge. As stated by the authors: "Despite [their] best effort, the model cannot escape from several fundamental deficiencies. One difficulty is with regard to the application of the one-dimensional Richard's equation with fine layers over large model grids of horizontal homogeneity. [Their] grid size of 2 km cannot differentiate hillslopes from first-order stream valleys, a fundamental scale of water movement on and near the land surface. This topographic gradient from hilltops to valleys also underlies many observed systematic changes in soil and vegetation. Resolving fluxes at this scale over continental regions is crucial but yet infeasible. A second but related difficulty is the use of coarsely gridded global soil maps such as the FAO product, obtained from agricultural surveys of topsoils ($\sim$1 m), for calculating water fluxes in very fine layers. [...] A third (and related to the second) difficulty is the complete lack of information on

the hydro-stratigraphy of the subsurface. Groundwater movement is controlled by the permeability structure of sediments and fractured rocks. Despite a century of aquifer characterization in many parts of the world, there remains a complete lack of basic data sets beyond the single-slope or single-aquifer scale, such as the depth to the bedrock and the vertical structures of porosity and permeability. Large-scale land models must rely on assumptions such as exponential decay of permeability with depth, which is widely adopted but at the same time widely known to grossly misrepresent the real-world." (Miguez-Macho and Fan, 2012).

The aim of our proposed hypothesis of the forest reservoir is not to produce a definite explanation of the causes for a river basin to be regulated or unregulated. Instead, our aim is to present a sound scientific hypothesis that may be further tested and discussed. Miguez-Macho and Fan (2012) concluded that "[the limitations of their study] can only be addressed collectively and in time. The saving grace is that the land surface topography has an enormous power in driving the movement of water at and near the surface. As shown [...], by simply allowing the gravity-driven flow in the subsurface, and letting the water level difference to determine the groundwater-surface water exchange, one can gain important, albeit qualitative, insights on the likely hydrologic states and fluxes near the land surface." In the same spirit, we consider that the difficulties in explaining the regulation capacity of river basins and its potential dependence on forest cover can only be addressed collectively and in time. The results and hypothesis presented in our paper are intended to be a contribution in this direction.

We foresee a potential danger in the assumption that the regulation capacity of river basins depends on geomorphological and geological processes with land cover (forest cover in the Amazon) playing a negligible role. Under this assumption, land cover change (e.g. vegetation change implying forest loss) would not change the capacity of river basins to regulate river flows. This is in contrast with many studies that show important land cover change effects on river flow regimes [References in the paper].

Our idea is not that the role of land cover (particularly forest in the Amazon) in river

flows regulation is more or less relevant than the role of geological and geomorphological processes, but that the role of land cover is not negligible, especially in the context of rapid global change. We agree that geological and geomorphological processes are first order drivers of basin's hydrological functioning (lines 15-25 p. 10). However, we want to emphasize that there is a fundamental difference between geological and geomorphological attributes and land cover-related biophysical properties. This difference relates to their sensitivity to global change.

In our response to the following comment we include a conceptual example to consider the idea that the regulatory capacity of a river basin is mostly controlled by geological and geomorphological processes, with land cover change-related processes being negligible. This will further develop our answer to this comment.
* * *
Comment:

4) Finally, the authors suggest a series of change due to the effect of land use and land cover changes on the ability of rivers to regulate the response. Although this is a conceptually sound hypothesis, demonstration based on river data observations, rather than modeling, have proven to be quite illusive in large basins. For instance, detailed analysis of trends on rivers (Marengo, 2009; Espinoza et al. 2009) showed that the trends detected are associated to interdecadal variability, instead of the potential effects of land use changes. Moreover, recent trends in the hydrological cycle of the Amazon have also been attributed to the warming of the Atlantic Ocean (Gloor et al. 2015) rather than local-scale changes. If we take into consideration that most of the ability to regulate river regimes is related to rainfall regimes and geological – geomorphological characteristics ,as demonstrated before, it might be challenging to disentangle LUCC effects from those major natural drivers.

Response:

To clarify our response to this and other comments, we propose the following concep-

Interactive
comment

tual example: compare the regulation capacity of two land cover scenarios for the same basin. In the first scenario the basin is predominantly covered by forests (the "Forested basin"). In the second scenario all the forests are lost (the "Deforested basin"). The only difference between scenarios is land cover. All other attributes of the basin, including geological and geomorphological properties that are independent of land cover, are the same in both scenarios. We assume that the rainfall regime is also the same in both scenarios (below we will revise this, because precipitation and land/forest cover are not independent). The question is whether the Forested and Deforested basins have the same capacity for regulating river flows. The answer would be yes, only if the effects of deforestation on river flows were negligible. Many observational and modelling studies have shown significant effects of deforestation on river flows [References in the paper], even under the assumption of precipitation invariance (this is assumed when modelling land cover scenarios with hydrological models that use precipitation as an input, but does not allow for vegetation feedbacks on precipitation).

The idea in our paper is that the regulation capacity would be significatively different between the Forested and Deforested basins, even if they had the same rainfall regime. Typical effects of tropical deforestation include reduction of surface permeability and infiltration capacity (due to, for instance, loss of deep and complex root systems, reduction of surface roughness associated with vegetation structural complexity, and increase of rainfall compaction effect), and increase of direct runoff (associated to a smoother, less permeable surface). A reduction of surface permeability and infiltration is consistent with reduced base flow and, therefore, reduced low flows. Similarly, an increase of direct runoff is consistent with increased floods. This exacerbation of extreme flows is consistent with a basin that has a lower capacity to store water (e.g. via infiltration) and to control its release (e.g. via base flow or direct runoff). Collectively, these effects are consistent with a basin (the Deforested basin) that has a lower (as compared to the Forested basin) capacity for dampening floods and amplifying low flows, i.e. a less regulated basin.

Regarding the potential influence of climate variability on regulation, it is important to recall our response to the second comment. In particular, that the regulation capacity of a basin system is a consequence of its internal dynamics rather than a result of the influence of external forcings. Our proposed hypothesis considers precipitation as a flux within the coupled-land atmosphere system of a river basin. We consider that precipitation is not independent of forest cover, especially in the Amazon. Many studies have shown that forest loss can alter precipitation regimes over the Amazon.

Our study does not aim to isolate the role of land cover (forests) from all other factors affecting river flows regulation. Our hypothesis of the forest reservoir is intended to describe mechanisms through which forests can exert an important influence on the capacity of river basins to regulate river flows. Advancing towards this understanding of the relation between forest cover and river flows regulation is crucial for water management- and land cover-related decisions.
* * *
Comment:

In conclusion, it is my opinion that the manuscript should go through a major revision. The authors need to explain better the role of LAI in the relationships they proposed (which I think is unnecessary). Regarding the forest reservoir hypothesis, the concept should be clearer if an example based on observations is brought into the manuscript. I presume that, along the "deforestation arc", several candidate basins undergoing through severe land use and land cover changes in the last decades can be found.

Response:

We are positive that we have successfully addressed the reviewer's concerns and are willing to provide further explanation if required. We appreciate the reviewers's suggestion about studying individual basins along the deforestation arc. However, this would not be feasible, as our proposed approach (based on scaling) requires multiple scale observations, such as those that we have already included in the analysis.
* * *
**References**

Espinoza Villar, J. C., Ronchail, J., Guyot, J. L., Cochonneau, G., Naziano, F., Lavado, W., ... & Vauchel, P. (2009). Spatio-temporal rainfall variability in the Amazon basin countries (Brazil, Peru, Bolivia, Colombia, and Ecuador). International Journal of Climatology, 29(11), 1574-1594.

Gupta, V. K., & Dawdy, D. R. (1995). Physical interpretations of regional variations in the scaling exponents of flood quantiles. Hydrological Processes, 9(3‐4), 347-361.

Miguez-Macho, G., & Fan, Y. (2012). The role of groundwater in the Amazon water cycle: 1. Influence on seasonal streamflow, flooding and wetlands. Journal of Geophysical Research: Atmospheres, 117(D15).

Molinier, M., Guyot, J. L., De Oliveira, E., & Guimarães, V. (1996). Les regimes hydroiogiques de l'Amazone et de ses affluents. IAHS PUBLICATION, 209-222.

---

## Author Comment (AC2) · 27 Jul 2017

Comment:

This work by Salazar et al. is an interesting study and presents original ideas. The authors study scaling properties of river flows of the Amazon basin and its subbasins. Identifying whether a basin attenuates or amplifies extremes in the flow regime, they propose that (Amazonian) river basins can go through tipping points of river flow regulation if forest loss exceeds a critical level.

Despite the interesting features of this study, I have a number of concerns that leave me yet unconvinced of some of the interpretations and conclusions drawn from them. These issues need to be addressed in a major revision before I can recommend publi-

cation of this manuscript.

Response:

We thank the reviewer for his/her comments. Specific answers to each comment are provided below.
* * *
Comment:

The authors hypothesize that, generally, river flows in Amazonian basins are regulated by the forests, meaning that extreme lows and highs in flows are attenuated by the forest. Mainly forest-induced precipitation recycling would be responsible for this attenuation. Indeed, this provides a positive feedback between the land/biosphere and rainfall, and such positive feedbacks are necessary for tipping points to occur (Van Nes et al. 2016, Trends in Ecology & Evolution 31:902-904). It should, however, be made more clear how land-atmosphere interactions cause both higher minima and lower maxima in river flows. Moisture recycling also has a typical spatial scale and direction. How does this affect the regulation of river flows and could the finding that the Tapajos is unregulated be an artefact of its size (and possibly shape)?

Response:

The proposed land-atmosphere mechanisms that lead to potential loss of streamflow regulation with forest loss in the Amazon are highlighted in Figure 6. In this figure, we indicate how forest loss reduces both infiltration and evapotranspiration fluxes, which result in decreased soil and atmospheric water storage. These reductions in infiltration and evapotranspiration are compensated by increases in direct runoff, which in turn result in increased floods (Figure 6a). As soil water storage is decreased in response to forest loss, base flow (occurring in the dry season) that depends on sub-surface runoff and surface-groundwater interactions is likely decreased (Figure 6b). In addition, reduced evapotranspiration can lead to a reduction of precipitation, and lengthening of the dry season, particularly in the Amazon basin, where precipitation recycling is a dominant climatic feature. These land-atmosphere mechanisms effects combined, potentially amplify the difference between low flows and floods, leading to loss of hydrologic regulation. To improve clarity, we propose a revised description for Figure 6 (which can be read in the response to next comment in this document).

In this paper we propose an approach for assessing hydrologic regulation based on the scaling properties of river flows. When applied to the Amazon tributaries, this approach allows the identification of different levels of regulation, including an unregulated basin such as the Tapajos. Our approach for assessing hydrologic regulation depends solely on the basin's scaling properties and, therefore, depends on river flow observations (as explained in Section 2 of the paper). However, the processes that result in specific scaling properties of each basin can be manifold and are related, generally, to the basin's biophysical attributes. We propose that forest-loss in the Amazon can affect river flow regimes in ways that lead to loss of regulation, which is then indicated by the basin's scaling properties (Section 3), such as in the Tapajos basin.
* * *
Comment:

Key for understanding the feedbacks in the system should be figure 6. However, it rather confused me, for the signs of the arrows do not seem to represent the sign of individual interactions: for example, evapotranspiration does not decrease (as is indicated now), but increase atmospheric water storage. And how could atmospheric water storage increase direct runoff? Both the figure and the text should be revised to guide the reader more to understand the core of the idea that is proposed.

Response:

Signs in Figure 6 should not be interpreted independently, but rather as part of a story that begins with forest loss. For instance, to clarify the reviewer's example: the decrease in atmospheric water storage results from a decrease in evapotranspiration

produced by the loss of forest. To avoid potential confusion, we will describe Figure 6 in the text as follows:

"Increased forest loss results in decreased Evapotranspiration (ET), related to loss of leaf area and root depth. As a consequence of decreased land-atmosphere water flux (ET), atmospheric water storage and precipitation recycling are reduced. Following general mass conservation principles in the long-term water balance for a basin (P=ET+R), when ET is reduced, R (direct runoff) increases. Increased forest loss can also reduce infiltration, both through changes in soil properties, as well as a consequence of increased runoff. Lower infiltration leads to decreased soil water storage, which feedbacks into a further reduction of ET. Overall, the combined effects of reductions in ET, soil water storage and increased runoff result in increased floods (Figure 6a).

Decreased water storage resulting from increased direct runoff in the wet season, results in decreased baseflow in the dry season, which corresponds, generally to lower low flows. In addition, forest loss can also lead to reduced base flow through a lengthening of the dry season, and a reduction of precipitation (Figure 6b). Both of these effects have been previously related to deforestation in the Amazon."
* * *
Comment:

Furthermore, I am not convinced of the threshold of 60% tree cover below which river basins shift from regulated to unregulated. The evidence for this threshold is that the Tapajos, inferred to be the only unregulated subbasin in the Amazon, is also the only one with an average tree cover of below 60%. This correlation is too weak to draw the conclusion that this threshold exists, let alone that deforestation has caused the Tapajos to pass a tipping point.

Response:

We thank this comment as it allows us to distinguish between what we can conclude from our proposed approach and what we propose as a hypothetical explanation of our result. We can conclude that the river flow regime in the Tapajos is unregulated, based on the behavior of its scaling exponents for low flows and floods ($B\_L < B\_F$), following the theoretical framework developed in Section 2. This conclusion, based on river flow observations for multiple gauges within the basin, does not address the causes of such unregulation. We propose the Forest reservoir hypothesis (Section 4) as a potential explanation linking forest cover and river flow regulation, and provide a conceptual framework highlighting the mechanisms that could lead to such linkage (Figure 6).

Previous studies have highlighted the potential effects of losing approximately 40% of forest cover in the Amazon, particularly on atmospheric and other hydrologic processes. We highlight this forest cover threshold, as it coincides with the the amount of forest cover that separates regulated from unregulated basins in our study. However, our results do not allow us to conclude that this amount of forest cover is a critical threshold in the Amazon basin.
* * *
Comment:

The authors also relate the 60% threshold to its correspondence to the threshold that can separate forests and savannas as alternative stable states. However, the latter threshold applies at local scales instead of at basin scale. A basin-scale average tree cover does not provide information about how far from such a threshold a forest is in any particular location; having a larger extent of grasslands in a basin does not necessarily mean that the forests in the basin are closer to a threshold. Indeed, the southern subbasins have more naturally occurring savannas and therefore lower average (subbasin-sclae) tree cover. The presence of these savannas is a result of rainfall seasonality (Staver et al. 2011, Science 334:230-232), which, as pointed out by referee 1, itself affects the regulation of river flows.

Response:

Based on this and the previous comment, we have modified the last paragraph in the discussion to exclude the sentences that indicate the existence of forest-savanna alternative stable states, as our results do not refer to alternative ecosystem states but rather to river flow regulation states. The modified paragraph reads as follows:

"A critical implication of our forest reservoir concept is that forest loss can induce a transition from the regulated state to the unregulated state in the Amazonian river basins. The value of the forest cover fraction where the inequality reverses from $\beta L > \beta M > \beta F$ (regulated state) to $\beta L < \beta M < \beta F$ (unregulated state) is $\sim 0.60$ (Fig. 5a), equivalent to $\sim 40\%$ deforested area in a river basin. This value coincides with previous studies suggesting that forest loss beyond $\sim 30$–$50\%$ constitute a critical threshold in the Amazon beyond which rainfall is substantially reduced and a shift in the biosphere-atmosphere equilibrium can occur (Boers et al., 2017; Lawrence and Vandecar, 2015; Hirota et al., 2011; Sampaio et al., 2007). Our empirical findings, as well as the forest reservoir concept, indicate that presence and absence of tropical forest cover is concurrent with the regulated and unregulated states, respectively: the Tapajos and Madeira are the less regulated basins and also the ones with the lowest forest cover in the region."

—————————————————————————————————————————————

Comment:

If the hypothesis that river flow regulation can pass tipping points holds, what would be the concrete consequences of such transitions? Obviously the limit case of infinitely high and low river flows will not be reached, so how do the authors see the future of the Tapajos and other basins if land use change continues? The paper lacks explicitness in this sense, which will leave readers like myself to question the validity of the forest reservoir concept.

Response:

We agree with the reviewer that the case of infinitely high and low flows will not be reached, as this is a mathematical solution for our proposed theoretical framework. We propose that the physically-feasible limits for low flows and for floods are, respectively, zero and the value of precipitation. In the case of low flows, when the "forest reservoir" is empty (i.e. no water storage in the soil) and there is no precipitation, base flow tends to zero. This kind of behavior is common in water-limited basins. In the case of floods, when the forest reservoir is limited in its storage capacity, almost all precipitation becomes instant runoff. Both extremes have important ecological, economic and social implications.

It is not the purpose of this paper to produce future scenarios in the basins, as we have only used historic records to test our regulation hypothesis. However, if forest loss advances in any of the basins (but particularly in the Tapajos which is currently in the unregulated state), extreme river flows will likely become more extreme, and this can be exacerbated as the regulation capacity of the basin is further reduced. We recognize that forest loss is a factor affecting river flow regulation, but acknowledge that it is not the only factor potentially affecting regulation. For example, changes in precipitation associated with large scale changes in atmospheric circulation linked to climate change, will affect river flow regimes independent of the basin's regulatory capacity.
* * *
Comment:

Minor points:

In figure 5 the bar charts for tree cover are presented relative to a baseline of 60%, suggesting independent evidence for such a baseline, whereas the results in the figure itself are the evidence for a threshold of 60%. Please change to bar charts for tree cover starting at 0.

Response:
Agree, we will revise the figure as suggested.
* * *
Comment:

In figure 3e, the dots indicate that the exponents are significantly different. Yet, it is also said that it cannot be rejected that the exponents differ from 1. One of these statements must be wrong.

Response:

Agree, points will be removed in the revised version of Figure 3e.
* * *

---

## Author Response (AR1)

**Comment:**

**Editor Decision: Reconsider after major revisions (further review by Editor and Referees)** (08 Sep 2017) by Patricia Saco
Comments to the Author:
Dear Authors,

We have received two detailed and insightful referee letters. After my own assessment of the manuscript and looking at the referee's comments, I agree with both reviewers that the paper presents an interesting study of scaling characteristics of flow regulation, and an application to the classification of large Amazonian sub-catchments. I also agree that some aspects of the analysis need further analysis, particularly in the discussion, before consideration for publication.

Response:

Thank you. We gratefully acknowledge that yours and the reviewer's comments have helped us to improve our manuscript. The main new elements of the revised version are the following:

a)  The Discussion section (Section 4) has been extended. It now includes three subsections about the use of LA as the scale parameter (Section 4.1); an extended discussion of the forest reservoir hypothesis (Section 4.2); and an extended discussion about the potential forest loss critical threshold (Section 4.3). Figure 6 is new, and former Figure 6 is now Figure 7.

b)  The Supplementary information has been extended to include results of the scaling analysis when using A (the traditional approach) as the scale parameter. We show that our main conclusions are consistent between both scaling models (using LA or A). These results are discussed in the main text. New elements include Figures S7 to S9 and Tables S9 to S15.
* * *
**Comment:**

The main points raised by the reviewers that need to be incorporated in a revised version of the manuscript are:

1) As mentioned by reviewer #1 Decreased water storage in the atmosphere, produces decreased precipitation (P is decreasing, so the sum of ET+R is not constant). It is difficult to understand that runoff can increase, and this is the idea of why there could be a shift from a wet to a dry state at longer timescales as suggested in other studies of precipitation recycling. It is clear that runoff can increase as a result of decreased evapotranspiration, but perhaps you need to clearly identify differences in response times and consequences for short and long time scales. This is still unclear in your current responses to the referee's comments. It is also important to consider that, as mentioned by reviewer #1, moisture recycling has characteristic spatial scales. Please revise figure #6 and its interpretation accordingly. Therefore, though the link between reduced evapotranspiration and increased discharge is clear, the link to precipitation recycling is not. This has important implications for the discussion, as it emphasizes precipitation recycling.

Response:

Sections 4.2. and 4.3 now include an extended discussion of our forest reservoir (FR) hypothesis (we are now using "hypothesis" instead of "concept"). Here we summarize some important ideas, more details are given in the revised version.

We agree that P may change as a result of external forcings related to, e.g., climate change and variability. However, the FR hypothesis does not require P to be constant or independent of external forcings. New Figure 6 (which shows the FR control volume) and Equations 8 to 10 help to clarify this. The revised version includes the following clarifications:

c) "P (precipitation), E (evapotranspiration) and I (infiltration) are not external fluxes but components of complex land-atmosphere interactions (e.g. precipitation recycling) that occur within the system and, therefore, are fundamental to the mechanisms that can explain the capacity of a basin system for regulating river flows. Although external forcings (e.g. climate change or variability effects) do affect the response of the system (R is not independent of Q), the capacity for regulating river flows can only be a consequence of the system's internal dynamics. Otherwise, if the response of a system simply follows external forcings (if R were entirely governed by Q), then there would be no capacity for regulation. Variations in the internal dynamics of water storage allow for the occurrence of different river flow regimes under the same external forcings." (par 13-9, i.e. paragraph in page 13-starting at line 9).

d) Equations 8 to 10 show that, for any given external forcing (net atmospheric moisture convergence: Q) R can increase or decrease depending on how water storage changes within the system. The revised version includes the following: "The occurrence of floods or low flows is related, respectively, to the abundance or scarcity of water, which depend on external forcings that determine whether Q is large or small during any given time period (e.g. wet and dry seasons). Floods dampening depends on the capacity of the basin to retain water when Q is large (wet season), which implies increasing water storage, consistent with [Eq. 9]. Analogously, low flows amplification depends on the basin's capacity for releasing

previously-stored water when Q is small (dry season), therefore reducing water storage as described by [Eq. 10]" (par 13-21)

e) "The importance of forests for the system's internal dynamics of water storage is highlighted by their relation with precipitation. Precipitation is not entirely determined by external forcings nor independent of the presence of forests. If precipitation regimes were independent of forest-related processes, then those regimes should not significantly change in response to forest cover change." (par 14-1)

We have extended the discussion about the effect of forests on river flows. Of note is that there is not a single, globally-applicable response to how river flows change as a result of forest cover change. The revised version includes the following clarifications:

f) "Forest loss does not reduce or increase river flows in every basin at every temporal and spatial scale [References in the manuscript]. Fundamental reasons for this are that forests have an inherent capacity to either increase or decrease the water balance components, and that these effects have a complex and dynamic nature. For instance, forests can increase or decrease E via, respectively, opening or closing stomata, which is related to water availability: stomatal aperture tend to be increased during drought stress and decreased during excessive water stress [References in the manuscript]. Further, forest loss can significantly alter the hydraulic properties of soils, especially by reducing infiltrability [References in the manuscript]. Through these impacts, forest loss can alter all the water balance components in complex ways. If the effect of forest loss were always to reduce E (due to reduction of the cumulative leaf area) with no impact on P (as implicitly assumed in hydrological models that use P as a fixed input) nor on the hydraulic properties of soils and regulation capacity of the basin, then forest loss should be always associated with increased R and, therefore, increased floods and low flows. Likewise, if the effect of forest loss were always to increase E (related to, e.g. weaker stomatal regulation, disruption of below canopy shading and stability, and increased wind speed over the surface) with no other effects, then forest loss should always lead to reduced R and, therefore, reduced floods and low flows. In both cases, the effect of forest loss on extreme river flows would always be in the same direction. In contrast, the forest reservoir hypothesis considers that forest loss can have contrasting effects on low flows and floods, mainly because the production of these extreme flows is governed by different processes occurring during different seasons." (par 16-2)

g) "[...] forests have a strong potential to enhance the capacity of river basins for storing water and controlling its release, as well as for producing contrasting and time-variable (e.g. seasonally different) effects on the water balance components. These dual and dynamic effects are key for regulation since it requires opposite effects on low flows (amplification) and floods (dampening)." (par 11-33)

We agree that P recycling is not a dominant processes at all spatial and temporal scales in every basin of the world. However, there is evidence that P recycling is a crucial process in the hydrological cycle of the Amazon and neighbouring basins. Using new Figure 6 and Fig.

7, we have extended the discussion about the role of P recycling in regulating river flows. The revised version includes the following clarifications:

[revised manuscript text omitted]

2) As mentioned by Reviewer #2, and based on the results presented in your supplementary material, it is unclear what is the advantage of using LA instead of area for the scaling relations. Please explain and include this aspect in the discussion. At the moment, I don't see a clear discussion between differences, and the supplementary figures (that show the trends for A and LA, are not discussed). This is linked also to the last comment of this same reviewer that mentions that the role of LAI in the relationships proposed needs to be better explained. Please also consider including a discussion of the differences in results by accounting for LAI as opposed to using A (S=A instead of S=LA).

Response:

Section 3 and Supplementary information now include results of the scaling analysis when using A as the scale parameter (the traditional approach). We show that the main results of our study are consistent among the two scaling models (using either LA or A). New elements that are included in the revised version are:

k) Section 4.1: "The use of LA as scale parameter"

l) Last Section of the Supplementary Information: "Scaling results using A as the scale parameter". This Section includes new Figures S7 to S9 and Tables S9 to S15.

m) Table 1 now includes results from both scaling models.

The main ideas that are discussed in Section 4.1. are the following:

n) "Our general idea about the classification of river basins is independent of using LA as the scale parameter." (par 10-3)

o) "The main results of our study are consistent among the two scaling models" (par 10-11)

p) "The use of A as the scale parameter relies on the idea that it represents the horizontal area over which precipitation falls. Using LA is conceptually consistent with this same idea, because LA describes the area through which evapotranspiration is transferred to the atmosphere." (par 10-19)

q) "Using LA allows to explore the influence of a changing scale parameter. LA is much more sensitive to global change than A…" (par 10-27)
* * *
**Comment:**

3) As mentioned by reviewer #2 the 60% threshold is not clear, as having just one "unregulated basin" is not enough to identify a threshold or transition. Note that the modified text, in the response to the comment, is still not addressing this issue.

Response:

We agree that we there is not enough evidence to conclude that 60% forest cover is a critical threshold, and have clarified the conclusions accordingly. However, we maintain our conclusion about the existence of different regulation states and levels (e.g. Table 1) because it is based on the physical interpretation of the observed scaling properties, following the conceptual framework developed in Section 2. This conclusion neither requires that forests play an important role in regulation, nor ignores the potential role of other factors. The forest reservoir is a theoretical hypothesis (presented in Section 4.2.) which implies that the regulation capacity of a river basin can be importantly sensitive to forest cover change. From this perspective, we discuss, theoretically, how forest loss can cause a transition from the regulated to the unregulated state in a river basin, especially in the Amazon (Section 4.3.). The revised version includes the following:

r) "Our conclusion that the Madeira and Tapajos are the less regulated basins, with Tapajos being unregulated (Table 1), relies only on the observed values of the scaling exponents, following the theoretical framework developed in Section 2. Therefore, this conclusion does not ignore the important role of geological and geomorphological processes [References in the manuscript]. Depending on the case study, different levels of regulation or transitions between states could be attributed to different causes. The forest reservoir hypothesis provides a potential explanation linking forest cover and river flow regulation. The idea is not that the effect of land cover (particularly forest cover in the Amazon) on river flows regulation is stronger than any other effect (e.g. geological and geomorphological effects), but that the role of land cover is not negligible and critically important because of its sensitivity to global change, especially in a region such as the Amazon where forest ecosystems are highly threatened and forest-related precipitation recycling plays a major role

[References in the manuscript]. We foresee a potential danger in the assumption that the regulation capacity of river basins depends on geomorphological and geological processes with land cover playing a negligible role. Under this assumption, land cover change (e.g. forest loss) would not change the capacity of river basins to regulate river flows." (par 15-16)

s) "The forest reservoir hypothesis implies that the regulation capacity of a river basin can be importantly sensitive to forest cover change. The size of artificial reservoirs determines their regulatory capacity. Likewise, the regulatory capacity of the forest reservoir depends on its size, which is related to the extent of forest cover. This implies that forest loss weakens regulation. The lower levels of regulation in the Madeira and Tapajos river basins (Table 1) are consistent with a weaker forest reservoir (these two basins are the less forested ones, Fig. 5a), likely related to extensive forest loss that has occurred along the arc-of-deforestation [References in the manuscript]." (par 16-19)

t) "The identification of alternative regulation states from scaling properties in river basins (Section 2), together with the hypothesis that forest loss weakens the regulatory capacity, imply that forest loss can cause a transition from the regulated state to the unregulated state. This also implies that there is a forest cover critical threshold where the transition occurs. In our results, the forest cover fraction in the less regulated basins is ~0.60, while in the more regulated basins it is > 0.70 (Fig. 5a), which suggest a possible range for the critical threshold. Although more-detailed studies are essential to understand regulation dynamics in different regions, as well as to identify potential critical thresholds, our analysis shows that scaling patterns may be used to characterize regulation states and infer transitions in river basins. Such empirical approaches are essential [References in the manuscript] because it is becoming clear that accurate mechanistic models to predict critical thresholds (or tipping points) are currently beyond our reach [References in the manuscript], and the detection of early-warning signals for critical transitions in complex systems (e.g. river basins) remains a fundamental challenge in environmental science today [References in the manuscript]." (par 18-10)

u) "We have applied the proposed interpretation of river flows scaling properties to the Amazon river basin and found both the regulated (all except the Tapajos) and unregulated (the Tapajos) states among its main tributaries. Then we proposed the forest reservoir hypothesis to describe the natural capacity of river basins to regulate river flows through land-atmosphere interactions (mainly precipitation recycling) that depend strongly on the presence of forests, especially in the Amazon. A critical implication of this hypothesis is that forest loss can force the Amazonian river basins from regulated to unregulated states. This provides further evidence about the possible outcome of widespread forest loss in the Amazon, potentially involving forest loss critical thresholds, a matter of great uncertainty and concern [References in the manuscript]." (par 18-31)
* * *
**Comment:**

Please also consider addressing all the other comments by the reviewer's in the revised paper.

Response:

The revised version now considers the Editor's and Reviewers' comments, including:

v) Reviewer 2 suggested to remove some dots from Fig. 3. We did not remove such points but clarified the interpretation. The Legend now includes the following: "Dots over the bars indicate whether the scaling exponent is significantly different to 1 ($p<0.05$, the dot is not over 1) or not (the dot is over 1)."

w) Figure 5 was modified after suggestion by Reviewer 2: forest cover now starts at 0.

---

## Author Response (AR2)

**Comment:**

**Editor Decision: Publish subject to minor revisions (review by editor)** (31 Dec 2017) by Patricia Saco
Comments to the Author:
Dear authors,

I believe that the revised manuscript has greatly benefited from the review process. The paper will be ready for publication after answering and clarifying the discussion in response to the minor comment of reviewer #1.

Best Regards,
Patricia Saco

Response:

Thank you very much. We gratefully acknowledge that yours and the reviewer's comments have helped us to improve our manuscript. Below we answer the reviewers' comments.
* * *
**Comment:**

**Anonymous Referee #4**
For final publication, the manuscript should be accepted as is.
I have looked at the reviews and I think the authors have answered the reviewer's comments adequately. The manuscript is a meaningful contribution for the audience of HESS.

Response:

Thank you very much for your comments and positive recommendation.
* * *
**Comment:**

**Anonymous Referee #3**
In my opinion, the revised version of the manuscript benefitted a lot from previous referees' comments. I have a minor point which require a bit clarification.

Response:

Thank you very much for your comments and positive recommendation. We agree that the Editor's and reviewers' commment have helped us to improve our manuscript, and have included that in the acknowledgements.

I appreciate the authors' approach to relate discharge (Q) with Leaf Area (LA) which can be more realistic than Area, A due to its influence on evapotranspiration, infiltration etc. Again, as authors stated (P10.L9-10) that their intention is not to use LA as a scale parameter in any river basin, moreover, using a scale parameter in the Amazon. However, I may miss the point in the text here! My understanding, based on Figures S1-S6, replacing the LA term instead of A term in the power-law scale does not always improve R2 of the relationship. Moreover using the LA term in scaling may detoriate or have no impact on the R2 terms of the relationships. A bit more clarification is required at this point.

Response:

You are right that using LA as the scale parameter does not always improve R2 in the empirical power laws (as compared to the case of using A). Our intention is not to show that LA improves such statistics (values of R2 for the power laws relating Q and A are generally high, as expected), but to the discuss that using LA is possible and meaningful. Importantly, the main results of our study are statistically significant and consistent among the two scaling models. We have clarified this in the text (P10.L11-12): "Although using LA as the scale parameter does not always improve $R^2$ in the scaling power laws (Supplementary Figs. S1--S6), the main results of our study are statistically significant and consistent among the two scaling models:...":

MINOR POINTS:
P16. L6. Subject-verb agreement. …stomatal aperture tendS to….
SuppP1.L7-8. The locationS of the 85 gauges ARE indicated…..
SuppP1.L9. The lenghtS of the records …..

Response:

All minor points were corrected.